# Autoreactive T cells target peripheral nerves in Guillain–Barré syndrome

L. Súkeníková[1,11], A. Mallone[1,11], B. Schreiner[2,3], P. Ripellino[4,5], J. Nilsson[6], M. Stoffel[7,8], S. E. Ulbrich[9], F. Sallusto[1,10] & D. Latorre[1 ✉]

Guillain–Barré syndrome (GBS) is a rare heterogenous disorder of the peripheral nervous system, which is usually triggered by a preceding infection, and causes a potentially life-threatening progressive muscle weakness[1]. Although GBS is considered an autoimmune disease, the mechanisms that underlie its distinct clinical subtypes remain largely unknown. Here, by combining in vitro T cell screening, single-cell RNA sequencing and T cell receptor (TCR) sequencing, we identify autoreactive memory CD4[+] cells, that show a cytotoxic T helper 1 (T$_H$1)-like phenotype, and rare CD8[+] T cells that target myelin antigens of the peripheral nerves in patients with the demyelinating disease variant. We characterized more than 1,000 autoreactive single T cell clones, which revealed a polyclonal TCR repertoire, short CDR3β lengths, preferential HLA-DR restrictions and recognition of immunodominant epitopes. We found that autoreactive TCRβ clonotypes were expanded in the blood of the same patient at distinct disease stages and, notably, that they were shared in the blood and the cerebrospinal fluid across different patients with GBS, but not in control individuals. Finally, we identified myelin-reactive T cells in the nerve biopsy from one patient, which indicates that these cells contribute directly to disease pathophysiology. Collectively, our data provide clear evidence of autoreactive T cell immunity in a subset of patients with GBS, and open new perspectives in the field of inflammatory peripheral neuropathies, with potential impact for biomedical applications.

Guillain–Barré syndrome (GBS) is a rare and potentially life-threatening disease of the peripheral nervous system that results in rapidly progressive muscle weakness, the loss of tendon reflexes and, sometimes, respiratory failure and autonomic dysfunction[1]. The disease can show marked heterogeneity in its clinical phenotype, course and outcome. In 95% of patients, GBS manifests as a monophasic disorder characterized by an acute phase that develops within four weeks, followed by a recovery period that can last for years[1]. The different disease subtypes are classified according to the types of nerve fibre affected and the nature of nerve degeneration. Acute inflammatory demyelinating polyneuropathy (AIDP), the most common form of GBS in Europe and North America, involves primary injury at myelin sheaths and Schwann cell components, whereas acute motor axonal neuropathy (AMAN) affects the membranes of nerve axons in the nodes of Ranvier[1,2]. Respiratory tract infections or *Campylobacter jejuni*-associated gastroenteritis precede the onset of disease in most patients, and the incidence of GBS can increase during outbreaks of infectious diseases, as described for Zika[1,3]. More recently, a link between SARS-CoV-2 infection and GBS has been suggested, but this remains controversial[4,5]. Despite the proven beneficial effects of plasma exchange and intravenous immunoglobulin therapy, almost 20% of patients with GBS remain severely disabled, and nearly 5% die from respiratory problems[1].

Our understanding of the immune-mediated mechanisms that underlie the distinct disease subtypes remains limited. The disease pathogenesis is likely to be a consequence of an aberrant immune response triggered by environmental factors, and so far no consistent associations with certain human leukocyte antigen (HLA) class I or II alleles have been described[6–8]. In *C. jejuni*-associated AMAN, pathogenic autoantibodies directed against gangliosides—glycolipids of the peripheral nerves—are thought to mediate neuronal damage through molecular mimicry[9,10]. However, anti-ganglioside antibodies are absent in most patients with GBS, especially in individuals with the AIDP variant, suggesting that other immune-mediated mechanisms are involved. The central role of autoreactive T cells targeting myelin antigens that are exclusively expressed in peripheral nerves (PNS-myelin)—namely, peripheral myelin protein 0 (P0), peripheral myelin protein 2 (P2) and peripheral myelin protein 22 (PMP22)[11]—has been established in experimental autoimmune neuritis, the animal model of AIDP[1]. Further observations that describe the infiltration of T cells into nerves[12,13] and altered distributions of T cell subsets in the blood of patients with GBS[14–21] suggest that autoreactive T cells exist and contribute to the pathophysiology of the disease in humans. However, despite a few indications[22–24], this aspect remains mostly elusive.

[1]Institute of Microbiology, ETH Zurich, Zurich, Switzerland. [2]Department of Neurology, University Hospital Zurich, Zurich, Switzerland. [3]Institute of Experimental Immunology, University of Zurich, Zurich, Switzerland. [4]Department of Neurology, Neurocenter of Southern Switzerland EOC, Lugano, Switzerland. [5]Faculty of Biomedical Sciences, Università della Svizzera Italiana, Lugano, Switzerland. [6]Department of Immunology, University Hospital Zurich, Zurich, Switzerland. [7]Institute of Molecular Health Sciences, ETH Zurich, Zurich, Switzerland. [8]Medical Faculty, University of Zurich, Zurich, Switzerland. [9]Animal Physiology, Institute of Agricultural Sciences, ETH Zurich, Zurich, Switzerland. [10]Institute for Research in Biomedicine, Università della Svizzera Italiana, Bellinzona, Switzerland. [11]These authors contributed equally: L. Súkeníková, A. Mallone. ✉e-mail: latorred@ethz.ch

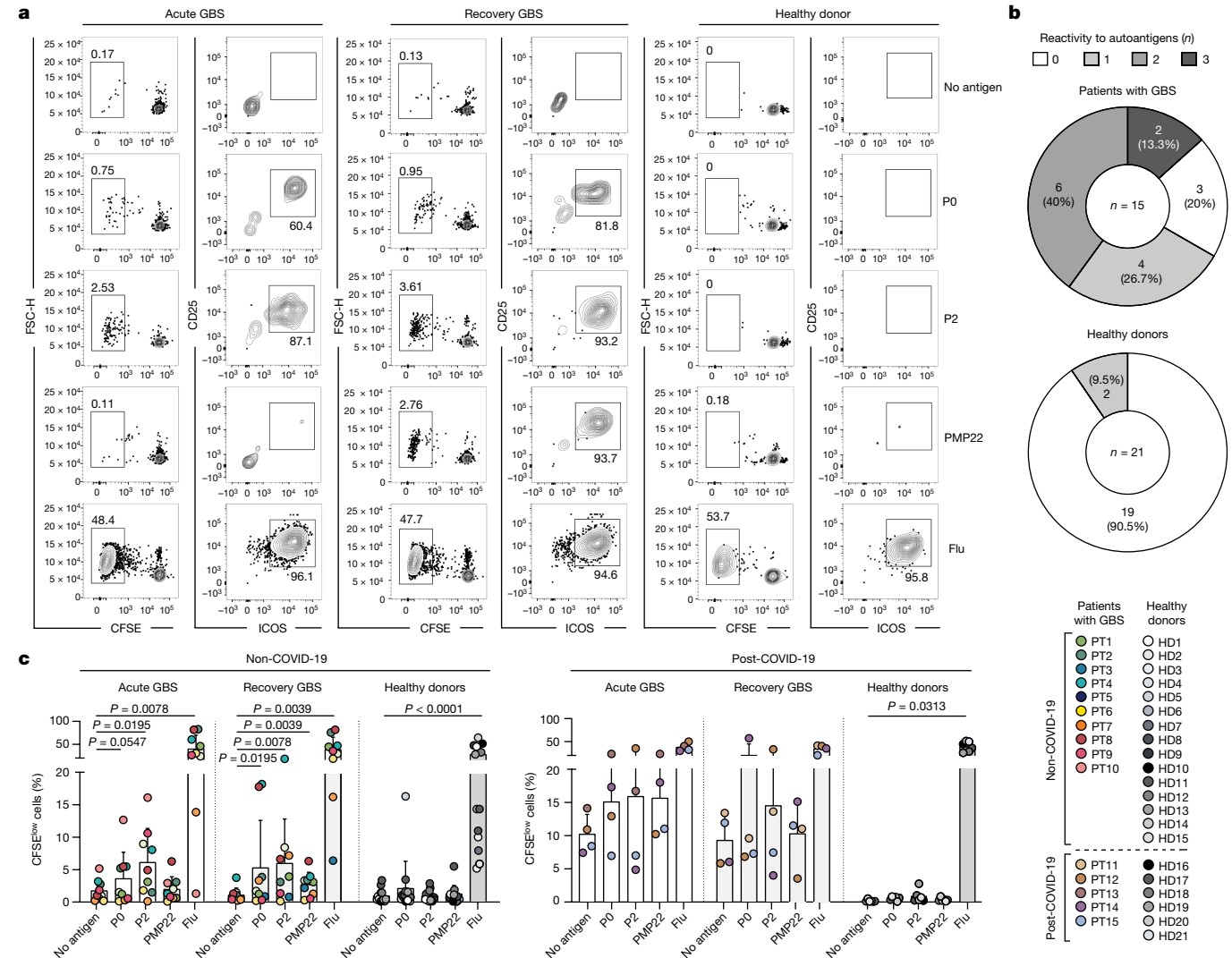

**Fig. 1 | Ex vivo stimulation of memory CD4+ T cells from the blood of patients with GBS and healthy donors.** Total memory CD4+ T cells from the blood of patients with GBS and healthy donors were labelled with CFSE and cultured with autologous monocytes in the presence or absence of PNS-myelin peptide pools (P0, P2 and PMP22) and influenza vaccine (Flu) as a positive control. **a**, CFSE profiles and dot plots of CD25 and ICOS expression of gated CFSE^low cells from one representative patient (PT1) in acute phase and in recovery phase and one healthy donor (HD4). **b**, Overview of the total number of screened patients with GBS and healthy donors who were reactive to either one, two, three or none of the three PNS-myelin autoantigens tested. A positive response was defined as a

stimulation index of 2 or higher and a Δ value of at least 1.5%. **c**, Scatter plot with pooled data from the indicated patients with GBS (*n* = 15 biologically independent samples; coloured dots) and healthy donors (*n* = 21 biologically independent samples; white and grey dots), shown as the percentage of proliferating CFSE^low cells. Patients with GBS are divided with respect to the disease phase (acute, *n* = 13; recovery, *n* = 14). Both patients with GBS and healthy donors are divided with respect to previous SARS-CoV-2 infection (non-COVID-19 GBS, *n* = 10; post-COVID-19 GBS, *n* = 5; non-COVID-19 HD, *n* = 15; post-COVID-19 HD, *n* = 6). Each dot represents an individual donor and bar height indicates mean and s.d. Data were analysed using two-tailed Wilcoxon matched-pairs signed rank test.

## Autoreactive T cells in patients with GBS

To investigate autoreactive T cell immunity in patients with GBS, we used an experimental approach that combines in vitro screening, single-cell RNA sequencing (scRNA-seq), the generation of single T cell clones and TCR sequencing (Extended Data Fig. 1a). The in vitro screening was performed on total memory CD4+ and CD8+ T cells from the matched blood samples from the acute and recovery stages of the disease of 15 patients with AIDP who had distinct potential infection triggers, including SARS-CoV-2 (non-COVID-19 and post-COVID-19 GBS) (Extended Data Tables 1 and 2)). As controls, we obtained blood samples from patients with AMAN (*n* = 4), patients with genetic demyelinating Charcot–Marie–Tooth disease (CMT) type 1 (CMT1; *n* = 5) (Extended Data Table 1) and healthy donors, some of whom with prior SARS-CoV-2 infection (non-COVID-19 HD, *n* = 15; post-COVID-19 HD, *n* = 6). In brief,

T cell populations were sorted by fluorescence-activated cell sorting (FACS) according to the gating strategy shown in Extended Data Fig. 1b, labelled with carboxyfluorescein succinimidyl ester (CFSE) and co-cultured with autologous monocytes in the presence or absence of selected PNS-myelin antigens (P0, P2 and PMP22) or positive control antigens (influenza vaccine for CD4+ T cells; Epstein–Barr virus (EBV) or human cytomegalovirus (CMV) for CD8+ T cells). Self-reactive memory CD4+ T cells targeting one or more PNS-myelin antigens were identified in 12 out of 15 patients with GBS at different disease stages, but not in healthy donors (except for 2 out of 21) (Fig. 1a,b). Moreover, autoreactive T cells were absent in patients with AMAN and were detected in one out of five patients with CMT1 (Extended Data Fig. 2a). The autoreactive response was directed against one or two self-antigens in 10 out of 15 patients with GBS, whereas only two patients with GBS showed broad autoreactivity against all three PNS-antigens (Fig. 1b).

P2 was the immunodominant target, identified in ten patients at different disease stages, whereas P0 or PMP22 were recognized in six patients (Extended Data Fig. 2b). Notably, autoreactive CD4[+] T cells were detected in ten out of ten non-COVID-19 patients with GBS, but in only two out of five post-COVID-19 patients with GBS (Extended Data Fig. 2c,d). In non-COVID-19 patients with GBS, P2 and—to a lesser extent—P0 were the main self-antigens targeted during the acute phase of the disease, whereas the autoreactive CD4[+] T response was significantly increased against all three PNS-myelin antigens during the recovery phase (Fig. 1c). Conversely, CD4[+] T cells from post-COVID-19 patients with GBS showed high background proliferation in negative control cultures (no antigen). This was not observed in post-COVID-19 healthy donors ($n = 6$), and, although PNS-myelin-reactive T cells were identified in two out of five patients (Extended Data Fig. 2c,d), the response was not significant (Fig. 1c).

Self-reactive memory CD8[+] T cells were detected in only 5 out of 11 patients with AIDP and in 2 out of 17 healthy donors (Extended Data Fig. 3a–c and Extended Data Table 2). In line with observations in memory CD4[+] T cells, the autoreactive memory CD8[+] T cell response was mostly found in non-COVID-19 (four out of seven) rather than post-COVID-19 (one out of four) patients with GBS (Extended Data Fig. 3c).

Collectively, these data indicate that PNS-myelin-reactive memory CD4[+] and rare memory CD8[+] T cells are present in the blood of most patients with AIDP, but that these cells are uncommon in patients with AMAN or CMT1 disorders, and in healthy donors.

## Cytotoxic T_H1 signature of autoreactive T cells

To gain insights into the phenotype and TCR repertoire of autoreactive CD4[+] T cells in patients with AIDP, we combined in vitro stimulation with scRNA-seq and paired TCRα and TCRβ (TCRα/β) analysis. In brief, memory CD4[+] T cells from two patients (PT2 and PT16) were stimulated in vitro with either PNS-myelin or influenza antigens, as described above. At day 6, for each condition, antigen-reactive CFSE$^{low}$ and non-reactive CFSE$^{high}$ T cells were FACS-sorted and combined in a single tube for scRNA-seq analysis, which identified 1,980 cells in cultures stimulated with PNS-myelin antigens and 2,232 cells in cultures stimulated with influenza antigens. Unsupervised clustering of our scRNA-seq data revealed two distinct clusters: one characterized by the expression of proliferation and activation genes[25,26], consistent with an antigen-driven condition (antigen-reactive cells) (Supplementary Table 1), and the second comprising low expression of proliferation and activation markers, typical of non-reactive T cells (Fig. 2a). PNS-myelin- and influenza-reactive T cell clusters comprised 413 and 414 single T cells encompassing 209 and 242 single TCRα/β clonotypes, respectively (data not shown). A comparison of these two clusters revealed that they were similar in their high average expression of the T_H1 gene signature, and low expression of the T_H2 or T_H17 signatures (Fig. 2b and Supplementary Table 1). T_H1-associated genes were enriched only in the antigen-reactive cell clusters, whereas a T_H2-like signature was mainly found in non-reactive cells (Fig. 2c). Notably, PNS-myelin-reactive cells showed substantially higher expression levels of genes associated with cellular cytotoxicity than did influenza-reactive cells (Fig. 2d and Supplementary Table 1). Finally, gene set enrichment analysis confirmed the activation status of antigen-reactive T cells and identified higher enrichment scores for genes previously associated with autoimmune conditions in PNS-myelin-reactive T cells compared with influenza-reactive T cells, which, by contrast and as expected, showed high gene-expression profiles associated with influenza virus infection (Fig. 2e).

These findings reveal an unique phenotype of autoreactive T cells in patients with AIDP. This phenotype is characterized mainly by the expression of T_H1-like genes and cytotoxicity markers, as well as by the expression of genes that have previously been associated with autoimmunity.

## Characterization of autoreactive T cell clones

To further examine the autoreactive T cell response in patients with GBS, we generated PNS-myelin-reactive single T cell clones ($n = 1,048$; Supplementary Table 2) from CFSE$^{low}$CD25$^{high}$ICOS$^+$ T cell fractions from in vitro screenings, which were further characterized for their TCRβ sequences, HLA restriction and targeted epitopes. We obtained a total of 987 CD4[+] T cell clones from 13 patients with specificities against P0 ($n = 312$), P2 ($n = 520$) or PMP22 ($n = 155$) (Fig. 3a), as well as 55 CD8[+] T cell clones from 6 patients targeting P0 ($n = 8$), P2 ($n = 14$) or PMP22 ($n = 33$) (Extended Data Fig. 3e). Autoreactive CD4[+] T cell clones predominantly expressed the pro-inflammatory cytokines interferon-γ (IFNγ) and tumour necrosis factor (TNF), along with the cytotoxic markers granzymes A and B (Extended Data Fig. 4a,b), providing corroborating evidence at the protein level for our findings from the scRNA-seq analysis (Fig. 2). Moreover, we determined the TCRβ clonotype composition of the autoreactive T cell clones, identifying 54 P0-reactive, 88 P2-reactive and 27 PMP22-reactive unique clonotypes from 706 CD4[+] T cell clones (Fig. 3a), as well as 4 P0-reactive, 7 P2-reactive and 6 PMP22-reactive unique clonotypes from 41 CD8[+] T cell clones (Extended Data Fig. 3f). Of note, in a few cases, the same TCRβ clonotype showed reactivity against both P2 and P0 (CD4_19 and CD4_43) or PMP22 (CD4_136) (Supplementary Table 2). Furthermore, P0- and P2-reactive sister T cell clones carrying the same TCRβ clonotypes were isolated from the matched acute and recovery blood samples of four patients with GBS (Fig. 3b and Supplementary Table 2). Both CD4[+] and CD8[+] autoreactive T cell clones showed a polyclonal TCR repertoire, including a broad spectrum of TCR Vβ genes even in the same individual (Fig. 3c and Extended Data Figs. 3f and 4c).

We next compared the *TCRB* complementarity-determining region 3 (CDR3β) length of PNS-myelin-reactive clonotypes from CD4[+] T cells ($n = 166$) with the SARS-CoV-2-specific ones ($n = 92$) from post-COVID-19 patients with GBS or with those of microbe-reactive CFSE$^{low}$ fractions from healthy donors[27,28]. The analysis was also performed on total memory CD4[+] T cells obtained ex vivo from peripheral blood mononuclear cells (PBMCs) from patients with GBS, as well as from publicly available datasets from healthy donors or patients with other autoimmune disorders[27–29] (Supplementary Table 3). Notably, the CDR3β lengths of PNS-myelin-reactive T cells were shorter than those of virus- and bacteria-specific or total memory CD4[+] T cell counterparts—providing further support for the self-reactive nature of these cells, as previously reported[30,31] (Extended Data Fig. 4d).

We also characterized the HLA restriction of 110 P0-specific, 194 P2-specific and 83 PMP22-specific CD4[+] T cell clones from 13 patients with distinct HLA haplotypes (Supplementary Table 4), accounting respectively for 37, 64 and 19 TCRβ clonotypes. This showed a preferential HLA-DR restriction (85.8%, $n = 91$), with a minority of clonotypes being HLA-DP restricted (7.5%; $n = 8$) or HLA-DQ restricted (8.5%; $n = 9$) (Fig. 3d, e and Supplementary Table 2). Finally, we successfully mapped the epitope specificity of TCRβ clonotypes specific for P0 ($n = 26$) and P2 ($n = 47$) from nine patients, and PMP22 ($n = 20$) from five patients, which revealed the recognition of multiple sites, collectively spanning the whole length of the PNS-myelin sequences (Fig. 3f and Supplementary Table 2). However, certain regions emerged as immunodominant, being targeted by several clonotypes across the patients with AIDP. Specifically, 6 clonotypes from 4 out of 9 patients recognized the P0 191–205 amino acid region, whereas 17 clonotypes from 5 out of 9 patients targeted the P2 1–15 amino acids, and 8 clonotypes from 5 out of 5 patients targeted the PMP22 81–100 amino acid region. No distinctive patterns of epitope recognition were observed when clones were analysed in relation to previous viral infection triggers, such as varicella-zoster virus (VZV) (PT1), SARS-CoV-2 (PT12, PT13 and PT14) or CMV (PT2 and PT3) (Fig. 3f). When screened for cross-reactivity, none of the P0-specific or P2-specific clones ($n = 52$) from post-COVID-19 patients (PT12 and PT14) proliferated in response to SARS-CoV-2

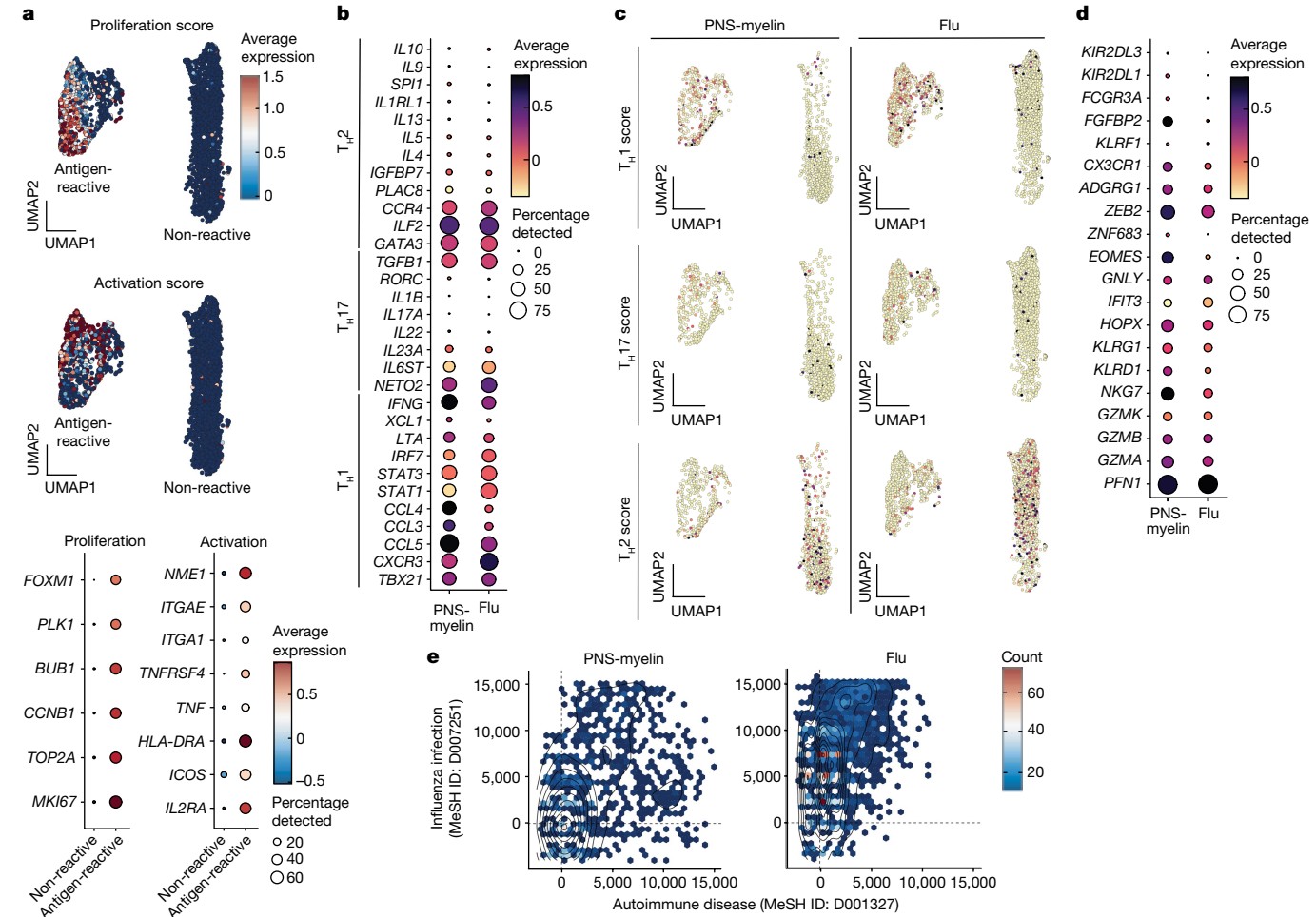

**Fig. 2 | scRNA-seq analysis of memory CD4⁺ T cells from patients with GBS.**
**a**, Uniform manifold approximation and projection (UMAP) and dot plots describing the average expression levels of activation and proliferation genes in CD4⁺ memory T cells from patients with GBS after in vitro stimulation with PNS-myelin antigens or influenza vaccine (Flu). Antigen-reactive cells exhibit a defined clustering and the upregulation of proliferation and activation gene signatures. **b**, Comparison of the average expression levels of genes from different T helper subsets (T$_H$1, T$_H$17 and T$_H$2 gene signatures) between PNS-myelin- and influenza-reactive CD4⁺ memory T cells. **c**, UMAP representing the average expression levels per cell of gene signatures of different T helper

subsets (T$_H$1, T$_H$17 and T$_h$2 scores) from PNS-myelin antigen and influenza conditions. **d**, Comparison of the average expression levels of cytotoxicity genes between PNS-myelin- and influenza-reactive T cells. **e**, Scatter plots with overlayed density plots comparing the distribution of enrichment scores of two distinct gene sets (autoimmune disease gene set, MeSH ID: D001327; influenza infection gene set, MeSH ID: D007251) across all single cells in PNS-myelin-reactive and influenza-reactive T cells. Data were analysed using two-tailed Wilcoxon matched-pairs signed rank test (P values are provided in Supplementary Table 1).

antigens (Extended Data Fig. 4e and Supplementary Table 2). However, when P2-specific (*n* = 14) or P0-specific (*n* = 18) clones from patients with prior CMV infection (PT2, *n* = 31; PT3, *n* = 1) were screened, most of them (*n* = 26) cross-reacted with CMV antigens (Extended Data Fig. 4f). Moreover, three out of six CMV-specific clones from PT2 proliferated in response to both P0 and P2 antigens (Extended Data Fig. 4f and Supplementary Table 2).

Overall, our data show that PNS-myelin-reactive T cells in patients with AIDP are mostly HLA-DR restricted, have a polyclonal TCRβ repertoire and short CDR3β lengths, and recognize multiple epitopes of the self-antigens, with some immunodominant regions being targeted across patients. Our findings also suggest that preceding infectious agents could be directly involved in establishing the disease by inducing self-reactive T cell immunity in a fraction of post-viral AIDP cases.

## TCRβ clonotypes in patients with GBS

We next studied the frequency of autoreactive T cells in the blood of patients with AIDP by high-throughput TCRβ sequencing. Specifically,

we compared the TCRβ sequences of our well-characterized autoreactive T cell clones with those of CD4⁺ memory T cells directly obtained ex vivo from PBMCs from the same patients (*n* = 7) (Extended Data Table 2). In several patients (PT1, PT2, PT4, PT5, PT7 and PT12), we identified TCRβ clonotypes corresponding to those of PNS-myelin specific T cell clones from the same patients (Fig. 4a). In line with the results obtained for single T cell clones (Fig. 3c), the same autoreactive clonotypes were found to be shared between the acute and the recovery samples of the same patient, and they showed variable frequencies in each donor (Fig. 3a). Autoreactive clonotypes were next cross-referenced against the TCRβ repertoire of memory CD4⁺ T cells ex vivo from PBMCs from patients with AIDP (*n* = 10) (Extended Data Table 2). This led to the identification of 18 PNS-myelin-reactive TCRβ clonotypes that were shared across several patients with AIDP (*n* = 6; Fig. 4b and Supplementary Table 2), and which were not detected in our previously published TCRβ datasets of memory CD4⁺ T cells from 15 healthy donors[27–29] (Fig. 4b). Specifically, two P0-specific clonotypes were detected, respectively, in six and five out of ten patients (60–50 %); two P2-specific clonotypes and one P0-specific clonotype

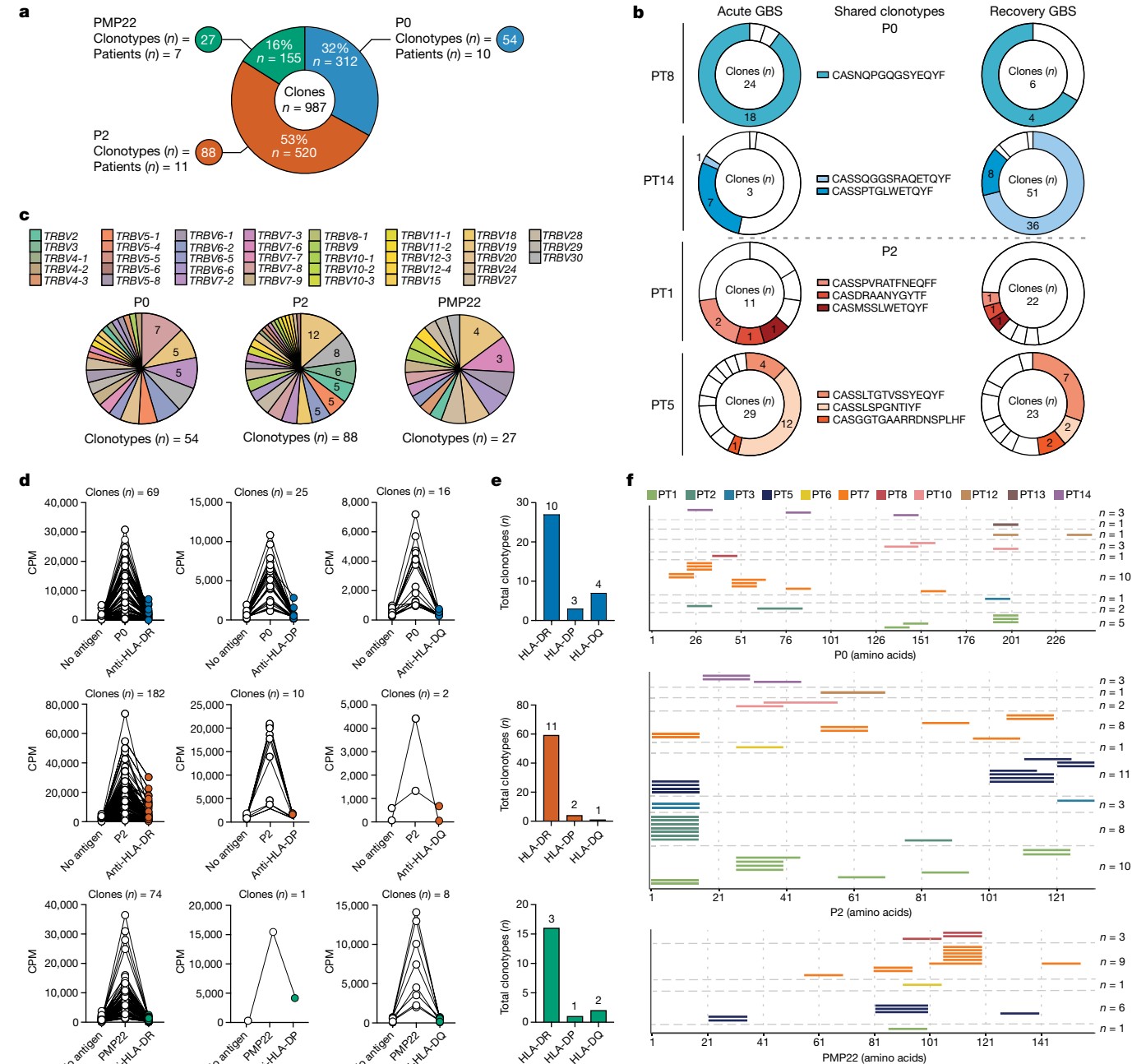

**Fig. 3 | Characterization of autoreactive CD4⁺ T cell clones from patients with GBS. a**, Total number of autoreactive CD4⁺ T cell clones and unique TCRβ clonotypes from patients with GBS. The number of patients from whom the autoreactive clones were obtained is indicated. **b**, Pie charts showing PNS-myelin-reactive clones and TCRβ clonotypes isolated from acute and recovery matched blood samples from the indicated patients with GBS (*n* = 4). Each slide represents a clonotype (PT8 acute *n* = 3; PT8 recovery *n* = 2; PT14 acute *n* = 5, PT14 recovery *n* = 5; PT1 acute *n* = 7, PT1 recovery *n* = 7; PT5 acute *n* = 11, PT5 recovery *n* = 11) and slice size is proportional to the number of isolated clones. The CDR3β amino acid sequences of clonotypes shared in acute and recovery stages are listed and highlighted in colour. The number of clones bearing the same CDR3β amino acid sequences is shown. **c**, Pie charts summarizing the TCR Vβ gene usage of PNS-myelin-specific clonotypes (P0 *n* = 54; P2 *n* = 88; PMP22 *n* = 27). Coloured slices in the charts represent different Vβ families and their size is proportional to the number of clonotypes. **d**, HLA restriction of autoreactive clones evaluated by measuring their proliferation against P0, P2 or PMP22 alone or in combination with HLA class II neutralizing antibodies (HLA-DR, HLA-DP or HLA-DQ). The number of tested clones is indicated. CPM, counts per minute. **e**, Summary results showing the number of autoreactive clonotypes (*y* axis) restricted to the indicated HLA class II molecules. The total number of patients with GBS from whom the autoreactive TCRβ clonotypes were isolated is shown above each bar. **f**, Graphical representation of epitope specificities of PNS-myelin-specific clonotypes. Each line represents the sequence recognized by a unique clonotype and its position on the *x* axis indicates the amino acid residues recognized.

were identified in four patients; and, finally, three P0-specific, P2-specific and five PMP22-specific clonotypes were shared across two or three patients (Fig. 4c). The cumulative frequency of the shared P0-specific clonotypes, ranging from $3.3 \times 10^{-5}$ to $1.4 \times 10^{-3}$ (median, $7.3 \times 10^{-5}$), was slightly higher than those of the shared P2-specific (range, $2.8 \times 10^{-5}$–$1.7 \times 10^{-4}$; median, $4 \times 10^{-5}$) and PMP22-specific (range, $6.5 \times 10^{-6}$–$9.3 \times 10^{-6}$; median, $8.1 \times 10^{-6}$) clonotypes in patients in the acute phase of the disease (Fig. 4d).

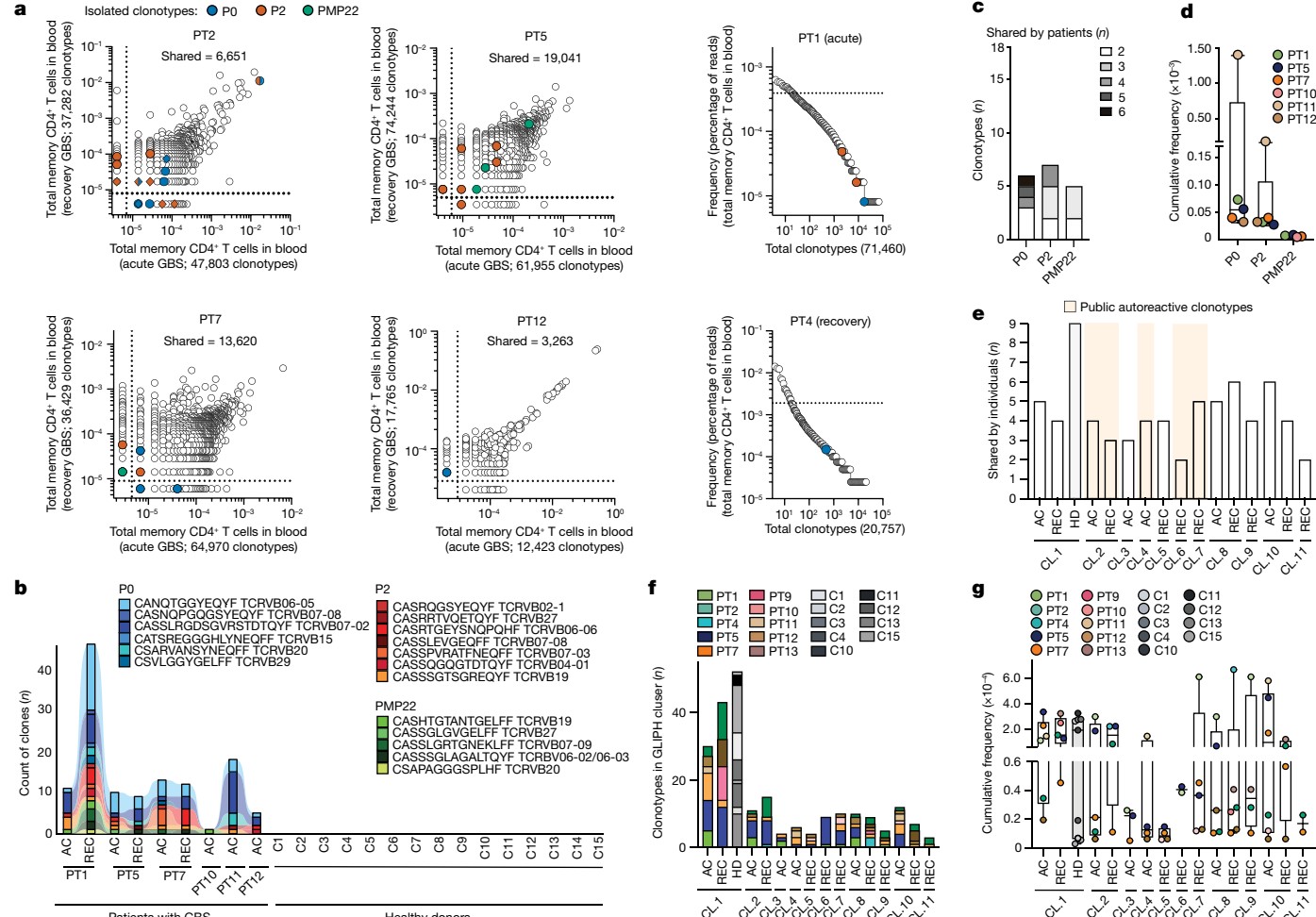

**Fig. 4 | Clonotypic analysis of autoreactive T cells in patients with GBS.**
**a**, Bulk TCRβ sequencing of total memory CD4+ T cells ex vivo from the blood of patients with GBS in the acute and recovery phases of the disease. The frequency distribution and numbers of total (*x* and *y* axes) and shared clonotypes are indicated. Coloured circles represent the frequency of autoreactive clonotypes (P0, blue; P2, orange; PMP22, green). Rhombus symbols indicate clonotypes cross-reactive with human CMV. Dotted lines indicate the frequency threshold of the top 20% expanded clonotypes. **b**, The number of clones (*y* axis) carrying shared autoreactive TCRβ clonotypes (*n* = 18, 6 P0-, 7 P2- and 5 PMP22-specific) in total memory CD4+ T cells from the blood of biologically unrelated patients with GBS at different disease stages (acute (AC) and recovery (REC)) but not in publicly available datasets from healthy donors (C1–C15). The TCRβ sequences and *TRBV* gene usage of public clonotypes are listed (P0, *n* = 6; P2, *n* = 7;

PMP22, *n* = 5). **c**,**d**, The number (**c**) and the cumulative frequency in the blood at acute disease phase (**d**) of autoreactive TCRβ clonotypes shared across patients with GBS (*n* = 18) is plotted (*n* = 6 biologically independent patient samples). Median values are shown, with boxes representing quartile values, whiskers the highest and lowest values and each dot a donor. **e**, Total number of individuals contributing to each GLIPH2 cluster (CL). Clusters including public autoreactive TCRβ clonotypes are highlighted in light orange. **f**,**g**, The identity of patients with GBS and healthy donors and their respective contribution to each GLIPH2 cluster in terms of TCRβ clonotype numbers (**f**) or cumulative frequency (**g**) are shown. Each dot represents a donor (*n* = 19 biologically independent samples; *n* = 10 patients with GBS (PT); *n* = 9 healthy donors (C)). Boxes are quartile values, whiskers represent the highest and lowest values and lines represent the median values.

We next performed an unbiased analysis of the TCRβ clonotypes in patients with AIDP by using the 'grouping of lymphocyte interactions by paratope hotspots' (GLIPH2) algorithm, which groups common clonotype specificities on the basis of local and global similarity[32]. We applied the GLIPH2 algorithm to the TCRβ repertoire of total memory CD4+ T cells from patients with GBS (*n* = 10) and antigen-reactive T cells obtained, respectively, by high-throughput sequencing and scRNA-seq analysis, as well as to published TCRβ datasets of memory CD4+ T cells from healthy donors[27–29] (*n* = 9), including our reference dataset of known PNS-myelin-specific clonotypes obtained from single T cell clones. The analysis identified a total of eleven TCRβ specificity clusters that included PNS-myelin-specific clonotypes on the basis of global (*n* = 10) or local (*n* = 1) similarity, each comprising at least four unique clonotypes from three or more individuals and exhibiting a significant final GLIPH2 score (Extended Data Fig. 5

and Supplementary Table 5). Notably, ten TCRβ specificity groups (clusters 2–11) were found exclusively in three or more patients with GBS and comprised P2-specific (*n* = 5) and P0-specific (*n* = 3) or PNS-myelin-reactive (*n* = 2) clonotypes from scRNA-seq, whereas cluster 1, including clonotypes with previously associated reactivities against self and viral antigens[33–36], was found to be shared in nine patients and nine healthy donors (Fig. 4e and Supplementary Table 5). Notably, clusters 8 and 9, comprising P0 and CMV cross-reactive clonotypes (Supplementary Table 2), were shared, respectively, in eight and four patients with GBS, but were absent in healthy donors. Moreover, four GLIPH2 clusters comprised autoreactive clonotypes that we identified as public in patients with AIDP (Fig. 4b,e). Each cluster comprised a variable number of clonotypes in each patient, accounting for a cumulative frequency ex vivo in the blood that ranged between $5.3 \times 10^{-6}$ and $6.7 \times 10^{-4}$ (Fig. 4f,g). Cluster one did

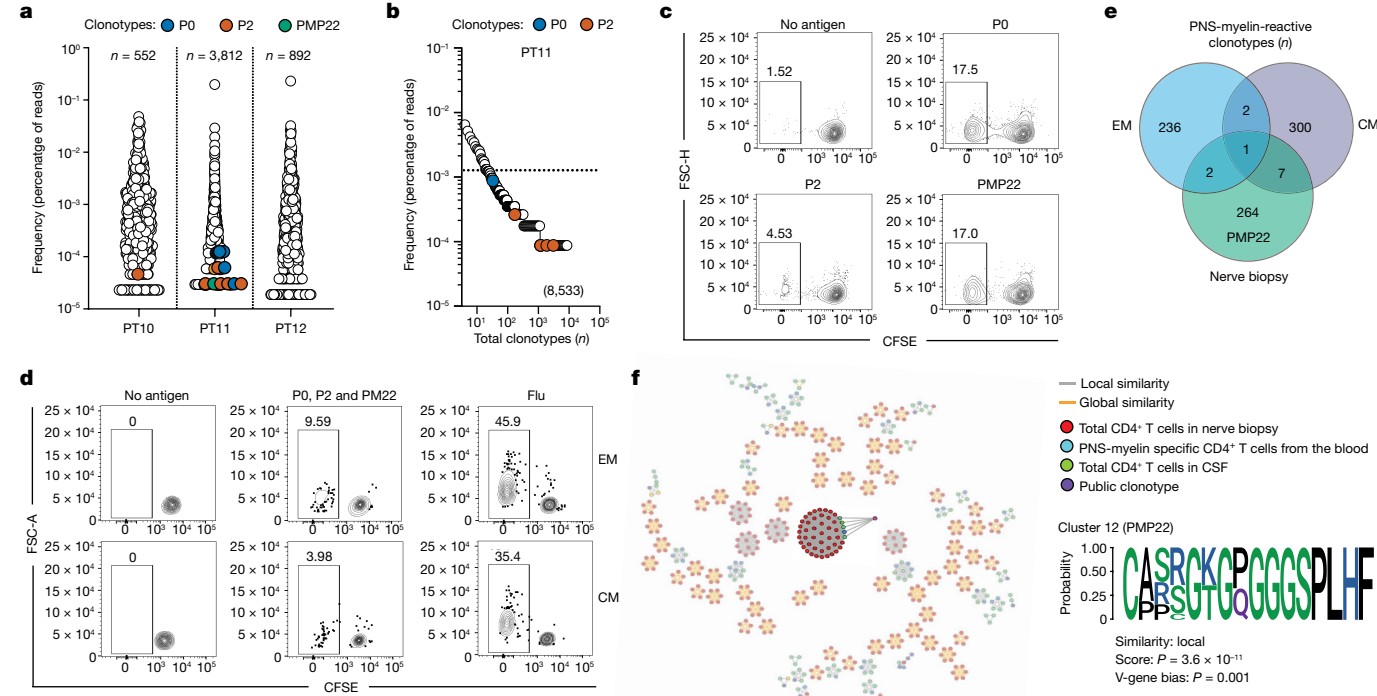

**Fig. 5 | Identification of autoreactive CD4⁺ T cells in the CSF and peripheral nerves of patients with GBS. a,b**, TCR Vβ sequencing was performed on in vitro expanded CD4⁺ T cells sorted from the CSF (**a**) or ex vivo sorted memory CD4⁺ T cells from the blood (**b**). TCRβ clonotype frequency distributions in the CSF (**a**) or the peripheral blood (**b**) of patients with GBS are shown as the percentage of reads. Coloured circles represent autoreactive TCRβ clonotypes (blue, P0-specific; orange, P2-specific; green, PMP22-specific). Values represent the total number of TCRβ clonotypes found in the CSF or blood. **c,d**, Total CD4⁺ T cells isolated from in vitro expanded T cells from a peripheral nerve biopsy (**c**) or EM and CM CD4⁺ T cells directly isolated from the blood (**d**) of one patient with GBS (PT16) were in vitro stimulated with autologous monocytes in the presence or absence of P0, P2 and PMP22 myelin peptide pools separately (**c**) or a mixture of P0, P2 and PMP22 peptide pools or influenza vaccine (Flu) as a positive control (**d**). Shown are the CFSE profiles of the total CD4⁺ T cells from the nerve biopsy (**c**) or of the EM and CM CD4⁺ T cells from the blood (**d**). **e**, Comparison of the TCRβ clonotype compositions of PNS-myelin-reactive CFSE^low cells in EM and CM CD4⁺ T cells from the blood and PMP22-myelin-reactive CFSE^low T cells from the nerve biopsy, identifying 10 unique TCRβ clonotypes shared between nerve tissue and blood. **f**, GLIPH2 graph showing identified paratope hotspots. Cluster 12 is highlighted and the specificity and consensus amino acid sequence are reported.

not show a specific enrichment in patients compared with healthy donors, in terms of either clonotype number or cumulative frequency (Fig. 4f,g).

Overall, these findings confirm the existence of expanded autoreactive memory CD4⁺ T cells in the blood of patients with AIDP at disease onset and recovery, and identify both public and private autoreactive TCRβ clonotypes with shared similarities and specificities across individuals with AIDP.

## Antigen recognition and HLA alleles

We next investigated the potential association between public autoreactive TCRβ sequences and HLA polymorphisms. Specifically, we examined the relationship between the HLA restriction of public autoreactive TCRβ clonotypes and the HLA class II alleles carried by patients with AIDP in whom the specific TCRβ clonotype was detected (Supplementary Table 4). Most of the public autoreactive clonotypes were HLA-DR restricted (n = 12), with only two being either HLA-DP or HLA-DQ restricted, respectively; the HLA restriction was not determined for four of the clonotypes (Extended Data Fig. 6a). Focusing on the HLA-DR-restricted ones, we did not identify any bias in *HLA-DRB1* allele sharing across patients, whereas the *HLA-DRB3* 02:02:01:02 allele was found to be shared by two patients for five out of six P2-specific and two out of five PMP22-specific clonotypes, and the *HLA-DRB4* 01:03:01:01 allele was shared by two patients for two out of six P2-specific clonotypes (Extended Data Fig. 6b). Along this line, when investigating the presence of an HLA polymorphism bias within

GLIPH2 clusters, we found that the clusters 2 and 3 had a significant HLA enrichment score driven by the *HLA-DRB3* 02 and *HLA-DRB1* 11 alleles, respectively (Supplementary Table 5). Finally, using NetM-HCIIpan[37], we performed binding-affinity prediction analysis of the cognate epitope for each public autoreactive clonotype in relation to the HLA alleles of the patients in whom that clonotype was identified. This analysis revealed that distinct HLA class II alleles were predicted to bind to the peptides within a similar range of affinities (Extended Data Fig. 6c).

These data indicate that there is relatively broad variability in antigen display across distinct HLA alleles, which might explain the lack of consistent association with definite HLA class II variants in patients with GBS[6–8].

## Autoreactivity in CSF and peripheral nerves

To investigate autoreactive T cells in the proximity of tissue immunopathology, we obtained a sample of cerebrospinal fluid (CSF) from three patients with AIDP at disease onset (Extended Data Table 1). Intrathecal CD4⁺ T cells were enriched and further characterized for their clonotype composition by high-throughput TCRβ sequencing, leading to the detection of 500–4,000 clonotypes in different samples (Fig. 5a). Of note, we identified PNS-myelin-specific clonotypes in the CSF of two out of three patients; specifically, one P2-specific clonotype in PT10 as well as four P0-specific, five P2-specific and one PMP22-specific clonotypes in PT11. In PT11, six out of the eleven autoreactive clonotypes identified in the CSF were also found in their blood

(Fig. 5b) and, notably seven of them were among those described as public in patients with AIDP (Fig. 4b, Supplementary Table 2). Conversely, none of the PNS-myelin-specific clonotypes was detected in our published TCRβ dataset of intrathecal CD4+ cells from patients with narcolepsy[29].

Finally, we obtained a nerve biopsy from one patient with AIDP (PT16) at disease onset. After in vitro polyclonal expansion, nerve-infiltrating CD4+ T cells were analysed for their antigen specificity by in vitro screening, which revealed the existence of T cells specific for P0 and PMP22, and, to a lesser extent, P2 (Fig. 5c). In parallel, we investigated autoreactivity in central (CM) and effector memory (EM) CD4+ T cell populations from the blood of the same patient, showing the existence of PNS-myelin-reactive T cells in both subsets with a slight enrichment in the EM population (Fig. 5d). To study the relationship between autoreactive T cells in the blood and the nerve tissue, we determined the TCRβ clonotype of the CFSE[low] fractions from in vitro stimulation, obtaining 274 unique sequences in PMP22-reactive T cells from the nerve biopsy as well as 241 and 310 unique clonotypes, respectively, in PNS-myelin-reactive EM and CM CD4+ T cells from the blood. We identified ten PNS-myelin specific clonotypes shared between nerve-infiltrating and blood-circulating T cells. Notably, seven and two clonotypes were found in the CFSE[low] fractions from CM and EM cells, respectively, whereas one clonotype was identified in both fractions (Fig. 5e).

To further investigate the presence of autoreactive clonotypes in different body compartments, we used the GLIPH2 algorithm to study the TCRβ clonotype repertoires of nerve-infiltrating CD4+ T cells and PNS-myelin specific CD4+ T cells from the blood of the same patient (PT16), as well as of total CD4+ T cells from the CSF of patients with AIDP. The analysis included our reference dataset of known PNS-myelin-specific clonotypes of single T cell clones (Supplementary Table 2). We identified one GLIPH2 cluster comprising one PMP22-specific public clonotype (CD4_149, Supplementary Tables 2 and 5) and encompassing several clonotypes grouped by local similarity in nerve-infiltrating ($n = 43$) and blood-circulating memory CD4+ T cells from the same patient (PT16) as well as in CSF-derived CD4+ T cells from three different patients (PT10, $n = 1$; PT11, $n = 1$; PT12, $n = 1$) (Fig. 5f and Supplementary Table 5).

Altogether, these results provide evidence for the existence of PNS-myelin-reactive T cells in the affected nerve tissue and the CSF compartment, pointing to their potential involvement in AIDP immunopathology.

## Discussion

This study provides a systematic description of CD4+ and CD8+ T cells targeting P0, P2 and PMP22 myelin antigens in the blood, CSF and nerve tissue of a well-characterized group of patients with GBS who have the demyelinating AIDP variant. Autoreactive memory CD4+ T cells showed a pro-inflammatory cytotoxic $T_H1$-like phenotype and expressed genes previously associated with autoimmunity. In line with previous observations[23], such cells recognized mostly P2 and, to a lesser extent, P0 antigens in the acute disease stage, whereas they were broadly directed towards several PNS-myelin antigens during disease recovery. These PNS-myelin proteins are essential for maintaining compact myelin in the peripheral nerves[38], and act as targets of pathogenic T cells in experimental autoimmune neuritis[1]. Our findings also identify common self-epitopes targeted across patients with AIDP, which are known to have a key physiological role. For instance, a large portion of the CD4+ T cell response against P0 recognized its cytoplasmic 180–199 amino acid residues, which are crucial for myelin integrity[39], are affected by point mutations in patients with CMT[40–42] and are targeted by pathogenic T cells in a spontaneous mouse model of autoimmune peripheral polyneuropathy[43,44]. Although our sample size is small, our data indicate the absence of an autoreactive T cell response in patients with AMAN

at disease onset, suggesting distinct underlying immune mechanisms. Axonal disease variants such as AMAN and Miller Fisher are generally considered to be mediated by autoantibodies[9,10]; however, future investigations should delve deeper into this aspect, and analyse a larger cohort of patients.

Moreover, we describe a polyclonal autoreactive TCRβ repertoire in patients with AIDP, which contains CDR3β sequences that are shorter than those of microbe-specific or total memory CD4+ T cells. Short CDR3β lengths have been linked to degenerate peptide responses[45] and autoreactive T cell immunity[30,31], suggesting that this may be a general feature of human autoimmunity. Furthermore, in line with the assumption that clonotypes with short CDR3β lengths are more likely to be shared across individuals[46], we identified a high degree of sharing of both identical sequences and motif similarity in autoreactive clonotypes across patients with AIDP, pointing to the existence of public disease-associated TCRβ clonotypes. Notably, prediction analysis revealed a promiscuous binding of peptides with similar affinity by distinct HLA class II alleles in patients with AIDP who shared public autoreactive TCRβ sequences. This points to a relatively broad variability in antigen display and recognition, which might explain the lack of consistent disease association with defined HLA class II variants[6–8]. Whether these findings apply to a larger spectrum of GBS clinical subtypes, chronic inflammatory demyelinating polyneuropathy (CIDP) or other autoimmune neuropathies remains to be investigated.

Although molecular mimicry has been largely described for autoantibodies in *C. jejuni*-associated cases of AMAN, the mechanisms that underlie post-infectious AIDP are unclear[9,10]. Despite COVID-19-associated GBS showing a classical AIDP-like demyelinating phenotype, our data indicate that only a minor fraction of these patients have an autoreactive T cell response against PNS-myelin antigens, which does not show cross-reactivity to SARS-CoV-2 antigens. This suggests that other self-proteins[47,48] or immune-mediated mechanisms have a role in post-COVID-19 patients with GBS. In this regard, the high degree of T cell auto-proliferation capacity observed in such cases supports a potential bystander mechanism[49]. Nevertheless, in AIDP cases associated with primary CMV infection, we identified T cells cross-reactive between self and viral antigens, which, in some cases, recognized two distinct PNS-myelin antigens and CMV. Accordingly, clonotypes that target two distinct PNS-myelin antigens were identified in three different patients with AIDP, suggesting a TCRβ degeneracy. Future research should investigate this aspect, which could prove particularly relevant owing to the widely recognized post-infectious origin of GBS.

Overall, our findings suggest that certain viral infections induce the activation of cytotoxic PNS-myelin-reactive CD4+ T cells that infiltrate the peripheral nerves, resulting in local inflammation and the recruitment of other immune cells[12,13], with subsequent myelin destabilization, epitope spreading and broadening of the immune response towards additional self-antigens at later stages of disease. The activation of autoreactive T cells might be sustained locally by the recognition of self-antigens presented by resident or infiltrating macrophages[50], or by Schwann cells, which show an enhanced antigen-processing capacity and increased MHC class II expression under inflammatory conditions[51–53]. It is unclear at present whether in patients with AIDP autoantibodies that target PNS-myelin proteins exist and have a role in the disease and whether autoreactive CD4+ T cells may contribute by providing B cell help for antibody production[54–60].

In summary, our results provide a comprehensive description of autoreactive T cell immunity in patients with AIDP, and further increase our understanding of the basic mechanisms that underlie GBS immunopathology. Our findings could pave the way for new medical interventions at the onset of symptoms to prevent disease progression and subsequent morbidity and mortality.

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

# Methods

## Study participants

The study included 20 patients with GBS and 5 patients with CMT1 recruited from University Hospital Zurich and the Cantonal Hospital of Lugano (EOC), and 21 healthy donors obtained from the Swiss Blood Donation Center of Lugano ($n = 15$) and from the CoV-ETH study ($n = 6$). All participants provided written informed consent for participation in the study. The study was approved by the ethical committees of Zurich (NeuroMyoCyTOF study, BASEC-Nr: 2016-00929; CoV-ETH study, BASEC-Nr: 2020-00949) and Lugano (IGOS study, BASEC-Nr: 2018-01860). We included patients who were diagnosed with AIDP ($n = 16$), as well as patients with AMAN ($n = 4$, all associated with preceding gastroenteritis) or CMT1 ($n = 5$) on the basis of the criteria for GBS of the National Institute of Neurological Disorders and Stroke (NINDS)[61] (Extended Data Table 1). Specifically, we included ten patients with AIDP who were sampled before the outbreak of the COVID-19 pandemic (non-COVID-19 GBS), for whom the potential trigger was either unknown ($n = 6$) or associated with VZV ($n = 2$) or CMV ($n = 2$) in the two to three weeks before disease onset. We also included five patients with AIDP with SARS-CoV-2 infection as a preceding trigger (post-COVID-19 GBS, range 6–20 days after infection, $11 \pm 5.4$ (mean ± s.d.)) (Extended Data Table 1). All patients with AIDP included in the study were HLA-typed by high-resolution next-generation-sequencing-based typing at University Hospital Zurich (Supplementary Table 4). One patient (PT16) also suffered from Waldenström's macroglobulinaemia in 2013, for which he received allogeneic bone marrow transplantation owing to progression in 2021. The patient had herpes zoster before developing severe GBS disease in 2022 (more than one year after transplantation), and mild pre-existing axonal polyneuropathy due to chemotherapy was documented. Peripheral blood samples from patients with AIDP were collected both at acute phase (range 7–36 days from disease onset, $12.7 \pm 9.9$ (mean ± s.d.)) and/or at follow-up visits during the recovery stage (range 135–509 days from disease onset, $244.6 \pm 115.7$ (mean ± s.d.)) (Extended Data Table 1). When available, we also obtained CSF ($n = 3$) at disease onset. Moreover, a nerve biopsy was obtained from the left sural nerve from one patient (PT16).

## Peptides and antigens

Peptides were synthesized as crude material on a small scale (1 mg) by Pepscan. Peptides used in the study included 15-mers overlapping by 10 covering the entire sequence of P0 (UniProtKB: P25189-1, $n = 48$), P2 (UniProtKB: P02689, $n = 25$) and PMP22 (UniProtKB: Q6FH25, $n = 30$) as well as human CMV and EBV HLA class I peptides (122 peptides, 46 EBV and 76 CMV). In some experiments, we used the heat-inactivated human CMV strain VR 1814 (ref. 62) or peptide pools covering the entire sequence of SARS-CoV-2 proteins; namely, spike-domain S1 (UniProtKB: QHD43416.1, S325 and S536-S685 amino acid, pool S1(ΔRBD), 91 peptides), spike-domain RBD (UniProtKB: QHD43416.1, S316-S545 amino acid, 44 peptides), spike-domain S2 (UniProtKB: QHD43416.1, S676-S1273 amino acid, 118 peptides), nucleocapsid (UniProtKB: QHD43423.2, 82 peptides), membrane (UniProtKB: QHD43419.1, 43 peptides) and envelope (ENV; UniProtKB: QHD43418.1, 13 peptides). Seasonal influenza virus vaccine Influvac 2019/2020 was obtained from Mylan.

## Cell purification and sorting

PBMCs were isolated with Ficoll-Paque Plus (GE Healthcare). Monocytes were enriched by positive selection using CD14-coated microbeads (Miltenyi Biotec). From the CD14⁻ cell fraction, memory CD4⁺ and CD8⁺ total cells were sorted to over 98% purity on a FACSAria Fusion (BD) excluding CCR7⁺CD45RA⁺, CD25^bright, CD14⁺ and CD56⁺ cells as well as either CD8⁺ cells (for memory CD4⁺ T cell enrichment) or CD4⁺ cells (for memory CD8⁺ T cell enrichment), according to

the gating strategy shown in Extended Data Fig. 1b. The following fluorochrome-labelled mouse monoclonal antibodies were used for staining: CD4–PE/Dazzle 594 (1:500, clone RPA-T4), CD45RA–BV650 (1:500, clone HI100), CD8–APC Fire750 (1:80, clone RPA-T8) and CCR7–BV421 (1:80, clone G043H7) from BioLegend; CD14–PE–Cy5 (1:30, clone RMO52), CD25–PE–Cy5 (1:30, clone B1.49.9) and CD56–PE–Cy5 (1:30, clone N901) from Beckman Coulter; and CD19–FITC (1:20, clone HIB19) and CD25–PE (1:20, clone M-A251) from BD Biosciences. Cells were stained on ice for 15–20 min and sorted on a FACSAria Fusion (BD Biosciences). Within a few hours of sampling, the nerve biopsy sample was minced and then filtered through a 40-µm cell strainer to obtain a single-cell suspension. CSF samples (1–2 ml) were collected by lumbar puncture. Cells from the nerve biopsy or the CSF were stimulated polyclonally with 1 µg ml⁻¹ PHA (Remel) in the presence of irradiated (45 Gy) allogeneic feeder cells ($1 \times 10^5$ per well) and IL-2 (500 IU ml⁻¹) in a 96-well plate format, as previously described[29]. On day 15, expanded T cells were stained with CD3–BV785 (1:100, clone UCHT1) and CD4–PE/Dazzle 594 (1:500, clone RPA-T4) antibodies from BioLegend, and CD8–FITC (1:30, clone B9.11) and CD56–PE–Cy5 (1:30, clone N901) antibodies from Beckman Coulter, and CD3⁺CD4⁺CD8⁻CD56⁻ or CD3⁺CD8⁺CD4⁻CD56⁻ T cells were sorted on a FACSAria Fusion (BD Biosciences).

## In vitro stimulation of T cells

T cells were cultured in RPMI 1640 medium supplemented with 2 mM glutamine, 1% (v/v) non-essential amino acids, 1% (v/v) sodium pyruvate, penicillin (50 U ml⁻¹), streptomycin (50 µg ml⁻¹) (all from Invitrogen) and 5% heat-inactivated human serum (Swiss Red Cross). Ex vivo sorted memory CD4⁺ and CD8⁺ T cells or EM and CM memory CD4⁺ T cell subsets (PT16) from the blood as well as in vitro expanded and sorted CD4⁺ T cells from CSF or nerve biopsy were labelled with CFSE and cultured at a ratio of 2:1 with irradiated autologous monocytes untreated or pulsed for 1 h with selected peptide pools from P0, P2 and PMP22 (3 µg ml⁻¹ per peptide) or with control antigens Inflexal V (5 µg ml⁻¹) or EBV or CMV (1 µg ml⁻¹). After six days, cells were stained with antibodies to CD25–PE (1:20, clone M-A251) and ICOS–Pacific Blue (1:100, clone H4A3) from BioLegend. The T cell response was scored positive on the basis of a cut-off value of (i) a stimulation index ≥ 2 (% of CFSE^low cells with antigen and APC/% of CFSE^low cells with APC only and (ii) a Δ value ≥ 1.5% (% of CFSE^low cells with antigen and APC − % of CFSE^low cells with APC only). This threshold was chosen on the basis of previous observations made across multiple negative and positive samples assessed by ex vitro T cell stimulation techniques in a variety of donors with self-antigens[29]. The list of samples analysed ex vivo is reported in Extended Data Table 2.

## Isolation of autoreactive T cell clones

To isolate autoreactive T cell clones, CFSE^low CD25⁺ICOS⁺ T cells from ex vivo cultures were sorted and cloned by limiting dilution, as previously described[29]. T cell clones were analysed by stimulation with irradiated autologous B cells that were untreated or pulsed for 1 h with P0, P2 or PMP22 peptide pools (3 µg ml⁻¹ per peptide). To determine MHC restriction, the assay was performed in the absence or presence of blocking anti-MHC class II monoclonal antibody (10 µg ml⁻¹; anti-HLA-DR, clone L243; anti-HLA-DQ, clone SPVL3; anti-HLA-DP, clone B7/21). In the cross-reactivity experiments with SARS-CoV-2 or CMV antigens, T cell clones were stimulated with irradiated autologous B cells after 2–3 h of pulsing with P0, P2 or PMP22 peptide pools (3 µg ml⁻¹ per peptide) or SARS-CoV-2 peptide pools (2 µg ml⁻¹ per peptide) or the heat-inactivated human CMV strain VR 1814 (2.5 µg ml⁻¹) (ref. 62). Epitope mapping experiments were performed by stimulating of autoreactive T cell clones with irradiated autologous B cells after one hour of pulsing with single 15-mer overlapping peptides (3 µg ml⁻¹ per peptide) covering the whole P0, P2 or PMP22 protein lengths. In all experiments, proliferation was measured on day 3 after 16-h incubation with 1 µCi ml⁻¹

[methyl-$^3$H]-thymidine (Perkin Elmer). Cell lines were routinely tested to exclude mycoplasma contamination.

## Cytokine analysis

For the quantification of cytokine release by autoreactive T cell clones, cells were stimulated with irradiated autologous B cells, either untreated or exposed for 1 h to P0, P2 or PMP22 peptide pools (3 µg ml$^{-1}$ per peptide). Cytokines released in the 48-h culture supernatants were quantified by the LEGENDplex multiplex bead-based immunoassay, using the predefined Human T Helper Cytokine Panels Version 2 (BioLegend) according to the manufacturer's instructions. Data were acquired using the FACS LSR Fortessa (BD Biosciences) and analysed with the Data Analysis Software Suite for LEGENDplex (BioLegend).

For intracellular cytokine staining, autoreactive T cell clones were restimulated with phorbol-12-myristat-13-acetat (PMA) and ionomycin in the presence of brefeldin A (all from Sigma-Aldrich) for the last 2.5 h of culture. Cells were stained with LIVE/DEAD Fixable Aqua dye (Thermo Fisher Scientific) and then fixed and permeabilized with Cytofix/Cytoperm (BD Biosciences) according to the manufacturer's instructions. After fixation, cells were stained with anti-granzyme A (1:50, clone CB9), anti-granzyme B (1:50, clone QA18A28), anti-perforin (1:50, clone dG9), anti-TNF (1:160, clone MAb11), anti-IL-10 (1:50, clone JES3-9D7) and anti-IL-17A (1:400, clone BL168) all from Biolegend; anti-IFNγ (1:160, clone B27) and anti-IL-4 (1:100, clone MP4-25D2) from BD Biosciences; and anti-IL-22 (1:50, clone 22URTI, Thermo Fisher Scientific), conjugated with different fluorochromes. Cells were acquired on a FACS LSR Fortessa (BD Biosciences) using BD FACS Diva (v.9.0) and flow cytometry data were analysed with FlowJo v.10.8.1 software (FlowJo).

## scRNA-seq analysis

scRNA-seq analysis was performed on memory CD4$^+$ T cells from two patients with AIDP (PT2 and PT16) at day 6 after in vitro stimulation either with a mixture of P0, P2 and PMP22 antigens (PNS-myelin antigens) or with influenza vaccine (Flu). Cells were incubated with a unique oligonucleotide barcode conjugated to a human universal antibody (Sample Tag, BD Single-Cell Multiplexing Kits) for backtracking both the condition and the patient of origin of each cell. For each condition, antigen-reactive CFSE$^{low}$ and non-reactive-CFSE$^{high}$ T cells were FACS-sorted to retrieve the total cell numbers and later combined at a 1:1 ratio in single tubes for further processing using the BD Rhapsody Express Single-Cell analysis system. In brief, cells were labelled with viability dies following the manufacturer's instructions and loaded onto BD Rhapsody Cartridges. The cartridges were subsequently analysed in the BD Rhapsody Scanner to obtain an estimate of the total cells and to verify their viability. After single-cell capture with the gravity-based, beads-assisted microwell technology we amplified the whole transcriptome, the TCR library and the Sample Tag library according to the manufacturer's protocols. We sequenced the library at the Functional Genomic Center Zurich (FGCZ) using the Illumina NextSeq 500 System. In detail, we sequenced 20,000 reads per cell for the WTA libraries, 5,000 reads per cell for the TCR libraries and 1,000 reads per cell for the Sample tag libraries. We used the SevenBridges online platform to perform read alignment on the reference genome 'Homo_sapiens_GENCODE_GRCh38-p13_Release_37-2021-05-04' and to generate feature-barcoded matrices for downstream analysis. The computational analysis allowed us to assign patient, condition, TCR and whole transcriptome information to each single cell analysed. After quality control, which involved the filtering of low-quality cells and cell doublets or multiples, and cells with mitochondrial counts higher than 5%, we normalized the data and performed scaling, dimensionality reduction and clustering on the top 2,000 highly variable features in the dataset (Seurat v.4.9.9.9059). In total, we obtained 1,980 cells (PT2 acute, $n$ = 608; PT2 recovery, $n$ = 262; PT16 acute, $n$ = 1,110) from cultures stimulated with PNS-myelin antigens and 2,232 cells (PT2 acute, $n$ = 287;

PT2 recovery, $n$ = 224; PT16 acute, $n$ = 1,721) from cultures stimulated with influenza vaccine. We later allocated the cluster on the basis of the expression levels of activation and proliferation genes[25,26] to define antigen-specific and non-specific clusters. Antigen-reactive T cell clusters comprised 413 PNS-myelin-reactive (PT2 acute, $n$ = 181; PT2 recovery, $n$ = 40; PT16 acute, $n$ = 192) and 414 Flu-reactive single T cells (PT2 acute, $n$ = 148; PT2 recovery, $n$ = 149; PT16 acute, $n$ = 117) encompassing, respectively, 209 and 242 single TCRα/β clonotypes (data not shown). We combined the acute and recovery datasets and compared the expression levels of gene signatures previously associated with different T helper subsets and cellular cytotoxicity[63–69] (all with Seurat v.4.9.9.9059) and then performed gene set enrichment analysis using the software package 'escape' (v.1.10.0, https://github.com/ncborcherding/escape)[70] using R v.4.2.1.

## TCR Vβ sequencing

To determine the TCR Vβ sequences of autoreactive T cell clones, total cDNA was obtained from 10$^3$–10$^4$ cells and TCR sequencing was performed following an established protocol[29]. In brief, the reaction was carried out using HPLC-purified oligo dT(25) primers (Microsynth) and Maxima H Minus reverse transcriptase (Thermo Fisher Scientific), in a reaction mix containing 0.2% Triton, dNTPs and RNase inhibitor. Reactions were run with the following conditions: 50 °C × 60 min; 55 °C × 5 min. Five microlitres of cDNA was added to a PCR mix (final volume 25 µl) containing Q5 Hot Start High-Fidelity DNA Polymerase (New England Biolabs). TCR Vβ sequences were amplified using TCR Vβ-specific forward primer pools and reverse primers pairing to constant regions, as previously described[29]. Sequence amplifications were assessed through agarose gel electrophoresis. Successfully amplified fragments were sequenced by the Sanger method, and TCR Vβ sequence annotation was performed using the IMGT/V-QUEST algorithm[29].

Deep sequencing of TCR was performed on CD4$^+$ memory T cells sorted ex vivo from PMBCs or in vitro expanded and sorted CD4$^+$ T cells from CSF or nerve biopsy as well as on PNS-myelin reactive T cells enriched as CFSE$^{low}$ fractions from in vitro stimulation ($2.5 \times 10^5$–$5 \times 10^5$ cells). In brief, cells were washed in PBS and genomic DNA was extracted from the pellet using the QIAamp DNA Micro Kit (Qiagen), according to the manufacturer's instructions. Sequencing of TCR Vβ was performed by Adaptive Biotechnologies using the ImmunoSEQ assay, as described previously[29]. In brief, after a multiplex PCR reaction designed to target any CDR3 Vβ fragments, amplicons were sequenced using the Illumina HiSeq platform. Raw data consisting of all retrieved sequences of 87 nucleotides or corresponding amino acid sequences, and containing the CDR3 region, were exported and further processed. Each TCRβ clonotype was defined as the unique combination of nucleotide sequence; data processing was done using the productive frequency of templates provided by ImmunoSEQ Analyzer v.3.0 (http://www.immunoseq.com) and by R package immunarch V.0.9.0 (https://github.com/immunomind/immunarch).

Antigen-specific TCRβ clonotypes in each donor's repertoire were identified through bioidentity overlap, defined as identical identified V gene, amino acid sequence of the CDR3 β region and identified J gene. The samples analysed are listed in Extended Data Table 2. Cumulative frequencies of shared TCRβ clonotypes were calculated as the sum of frequencies of each TCRβ clonotype in the respective patient's TCR Vβ repertoire. CDR3β length was calculated on the total productive rearrangements from the ImmunoSEQ Analyzer v.3.0 or the IMGT/V-QUEST algorithm.

## GLIPH2 analysis

The GLIPH2 algorithm[32,71] from the HetzDra/turboGliph v.0.99.2 R package (https://github.com/HetzDra/turboGliph/) was used to identify lymphocyte interaction by paratope hotspots and predict specificity groups, herein referred as clusters, on the basis of global

or local similarity (convergence). TCR global convergence relies on the CDR3 hamming distance between TCRs; namely, the number of different amino acid residues within the CDR3 region amongst two TCRs with identical length and sharing the same Vβ segment. TCR local convergence relies instead on similarity based on shared CDR3 amino acid motifs (2mers, 3mers, 4mers and 5mers) within any given set of T cell receptors (>10× fold enrichment, probability < 0.001). Notably, TCRs are allowed to be assigned to multiple clusters if computed similar to one another. GLIPH2 scores result from a combination of probabilities of a set of features, which are then combined into a single score by conflation. Such features include global similarity probability, local motif probability, network size; enrichment of V gene in the cluster, enrichment of CDR3 length in the cluster, enrichment of clonal expansion in the cluster and enrichment of common HLA alleles among TCRs from donors contributing to the cluster. The GLIPH2 algorithm is trained by a reference dataset of 162,165 CDR3β sequences and the query sample size should be comparable to the size of the training set[32,71]. Therefore, we run multiple rounds of analyses on different groups of patients and cohorts. We performed the GLIPH2 analysis on the TCR Vβ repertoire of total memory CD4+ T cells from the blood of patients with GBS (n = 10), grouped by disease phase. In detail, we analysed seven samples from the acute phase (PT1, PT2, PT5, PT7, PT10, PT11, PT12; total: 82,826 TCR Vβ sequences) and eight samples from the recovery phase (PT1, PT2, PT4, PT5, PT7, PT9, PT12, PT13; total: 239,501 TCR Vβ sequences, two rounds). We also performed the analysis on CFSE[low] enriched fractions of PNS-myelin specific CD4+ T cells from PT16 (EM and CM T cell populations from the acute phase; total: 568 TCR Vβ sequences) and on a published TCR Vβ dataset of memory CD4+ T cells from healthy donors (n = 9; C5, C6, C7, C8, C9, C10, C12, C13, C15; total: 149,939 TCR Vβ sequences, two rounds)[27–29]. In addition, we applied the GLIPH2 analysis to the TCR Vβ repertoire of total CD4+ T cells expanded from the CSF of patients with GBS (n = 3; PT10, PT11, PT12; total: 2,525 TCR Vβ sequences) and from the nerve tissue of one patient with GBS (n = 1, PT16; total: 99 TCR Vβ sequences). The TCR Vβ sequences from PNS-myelin specific memory CD4+ T cells isolated from the blood after in vitro stimulation (PT16), and total CD4+ T cells from the CSF and nerve biopsy (PT16) were analysed together in one round of GLIPH2 computations. Finally, we conducted one further round of GLIPH2 analyses on the TCR Vβ repertoire retrieved from scRNA-seq experiments on two patients with GBS (PT2 and PT16; total: 1,733 unique TCR Vβ sequences). In detail, the analysis was conducted on 526 TCR Vβ sequences from antigen-specific and 1,207 TCR Vβ sequences from non-specific CD4+ memory T cells after six days of stimulation with PNS-myelin antigens or influenza vaccine. In each round of analysis, we included the autoreactive TCRβ clonotypes isolated from the blood of patients with GBS (n = 167, of which n = 18 were shared across the memory CD4+ T cells TCR Vβ repertoires of several patients). Clusters were considered of relevance if they included one autoreactive TCRβ clonotype of known specificity and were shared by multiple patients with GBS. If the same cluster could be identified in different rounds of GLIPH2 analysis amongst different groups (for example, GBS acute, GBS recovery or healthy donors), that cluster would be considered as one, but the identifier code would be maintained to preserve positional information (Supplementary Table 5).

### Prediction of binding affinity of self-epitopes to HLA class II alleles

Binding-affinity predictions between HLA alleles carried by patients with GBS and the PNS-myelin peptide of interest identified through epitope mapping were performed using the NetMHCIIpan-4.0 server provided by the DTU Health Tech Department of Health Technology[37]. In brief, the artificial neural networks are trained over half a million experimental measurements of binding affinity and eluted ligand mass spectrometry covering the human HLA-DR, HLA-DQ and HLA-DP. When instructed with information regarding the HLA subtype of interest and a peptide of choice (15-amino-acid peptides), it can forecast the likelihood of a peptide being naturally presented, its predicted affinity and, the likelihood of that peptide being presented as compared with a group of random peptides. From NetMHCIIpan-4.0, we extrapolated the binding affinity of the HLA alleles of each patient known to be carrying public TCRβ clonotypes versus the specific epitope recognized by those public TCRβ clonotypes.

### Reporting summary

Further information on research design is available in the Nature Portfolio Reporting Summary linked to this article.

### Data availability

Publicly available datasets included in the study are available through immuneACCESS (https://doi.org/10.21417/JSL2021S, https://doi.org/10.21417/AC2020EJI and https://doi.org/10.21417/B73H0P), VDJdb (https://vdjdb.cdr3.net/search), the Gene Expression Omnibus (https://www.ncbi.nlm.nih.gov/geo/query/acc.cgi; accession numbers GSE59114, GSE126030, GSE131935, GSE104024 and GSE193442) and the European Genome-phenome archive (https://ega-archive.org/; accession numbers EGAS00001003215 and EGAD00001005290). All data associated with this manuscript are available in the main text or its Supplementary Information, including the FACS data gating strategy. TCR Vβ sequences from samples listed in Extended Data Table 2 have been deposited in the immuneACCESS database (https://doi.org/10.21417/LS2023N). All further relevant data that support the findings of this study are available from the corresponding author upon reasonable request.

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

**Acknowledgements** We thank all patients for their participation in the study; the ETH Zurich Flow Cytometry Core Facility for cell sorting; F. Wagen (ETH Zurich) for the isolation of PBMCs from non-COVID-19 healthy donors; F. Mele (Institute for Research in Biomedicine, Bellinzona) and F. Ingelfinger (University of Zurich) for the initial isolation of PBMCs from patients with GBS; E. Edwards (Universitätsspital Bern) for initial experimental support; G. Lüders (University Hospital Zurich) for HLA typing; A. Can (University Hospital Zurich) for support with the acquisition of clinical data for patients with GBS in Zurich; F. Ingelfinger and B. Becher for their support in setting up the NeuroMyoCyTOF study ethical protocol; the IGOS Steering Committee for approving this study as an amendment to the main IGOS study; J. Goldhahn and team for contributing to the CoV-ETH study; and A. Lanzavecchia for helpful discussions. This work was sponsored by grants from ETH Zurich (ETH Career Seed Grant, SEED-02 19-1), the Swiss National Science Foundation (PRIMA grant PR00P3_185742), the Swiss Foundation for Research on Muscle Diseases (FSRMM) and the Elevation Research Grant from the GBS/CIDP Foundation International to D.L.; the GBS/CIDP Foundation International, the GBS/CIDP Initiative Schweiz, the Foundation for Progress in Neurology of Lausanne and the Baasch-Medicus Stiftung of Zurich to P.R.; the Theodor and Ida Herzog-Egli Foundation to B.S. and A. Can; the Neuromuscular Research Association Basel and F. Hoffmann-La Roche to B.S. and F. Ingelfinger; and the ETH Zurich Foundation to M.S. and S.E.U.; F.S. and the Institute for Research in Biomedicine are supported by the Helmut Horten Foundation.

**Author contributions** B.S. and P.R. contributed equally to this work for the second pair. D.L. acquired funding for immunological analyses and conceived and supervised the project. L.S. and A.M. performed experiments, analysed the data and prepared the figures with assistance from D.L. A.M. performed all bioinformatics analyses. B.S. and P.R. recruited participants, performed clinical evaluation and collected biological samples. J.N. supervised HLA typing. M.S. and S.E.U. recruited participants and collected biological samples for the CoV-ETH study. D.L supervised data analysis and wrote the original draft with the help of L.S. and A.M. All authors provided input and critical revision of the manuscript.

**Funding** Open access funding provided by Swiss Federal Institute of Technology Zurich.

**Competing interests** The authors declare no competing financial interests. P.R. is a member of the advisory boards for Roche, Biogen, Alexion and Argenx Pharmaceuticals, all of which are not connected to this project.

**Additional information**
**Correspondence and requests for materials** should be addressed to D. Latorre.

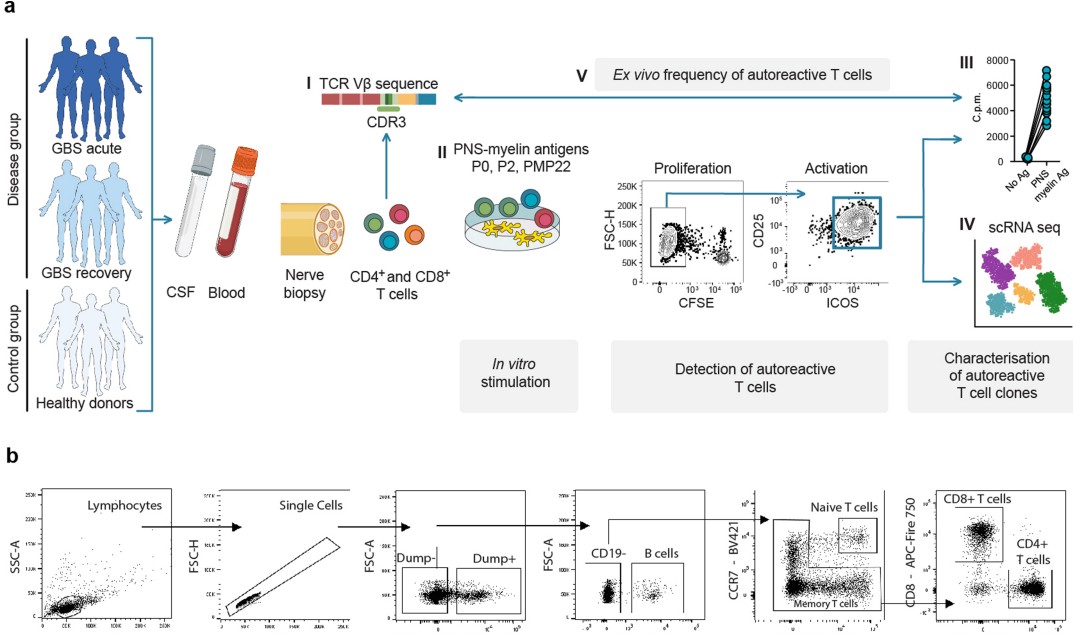

**Extended Data Fig. 1 | Experimental approach for studying autoreactive T cells in patients with GBS. a**, Peripheral blood from patients with GBS at acute phase and/or at follow-up visits during the recovery stage (Extended Data Table 1) as well as from healthy donors was collected. When available, we also obtained CSF ($n$ = 3) and nerve biopsy ($n$ = 1) samples at GBS disease onset. Memory CD4$^+$ and CD8$^+$ T cells ex vivo from blood or total CD4$^+$ and CD8$^+$ T cells, polyclonally expanded in vitro from CSF or nerve biopsy, were sorted by FACS cytometry. (I) A fraction of cells ($2 \times 10^5$–$2.5 \times 10^5$ cells) was analysed by high-throughput TCR Vβ sequencing to determine their repertoire composition. (II) The rest of the sorted T cells were in vitro screened for the presence of autoreactive T cells specific for P0, P2 and PMP22 myelin antigens of the peripheral nerves (PNS-myelin antigens). Autoreactive T cells were detected by CFSE dilution and expression of the activation markers CD25 and ICOS on day 6–7 by flow cytometry. When identified, autoreactive T cells were further studied at single-cell resolution through (III) the generation of single T cell clones by the characterization of their cytokine production, TCR Vβ sequences, MHC restriction and targeted epitope and (IV) scRNA-seq and paired Vα/β TCR analysis. (V) TCR Vβ repertoire of PNS-myelin-specific T cells was compared to the repertoire obtained from T cell populations directly isolated ex vivo from PBMCs or expanded from CSF or nerve biopsy to obtain information on their frequency in vivo in the blood and on the potential existence of autoreactive TCR clonotypes shared across patients with GBS. Human silhouettes, lab apparatus and nerves were created with BioRender.com. **b**, Gating strategy used to isolate ex vivo CD4$^+$ and CD8$^+$ memory T cells.

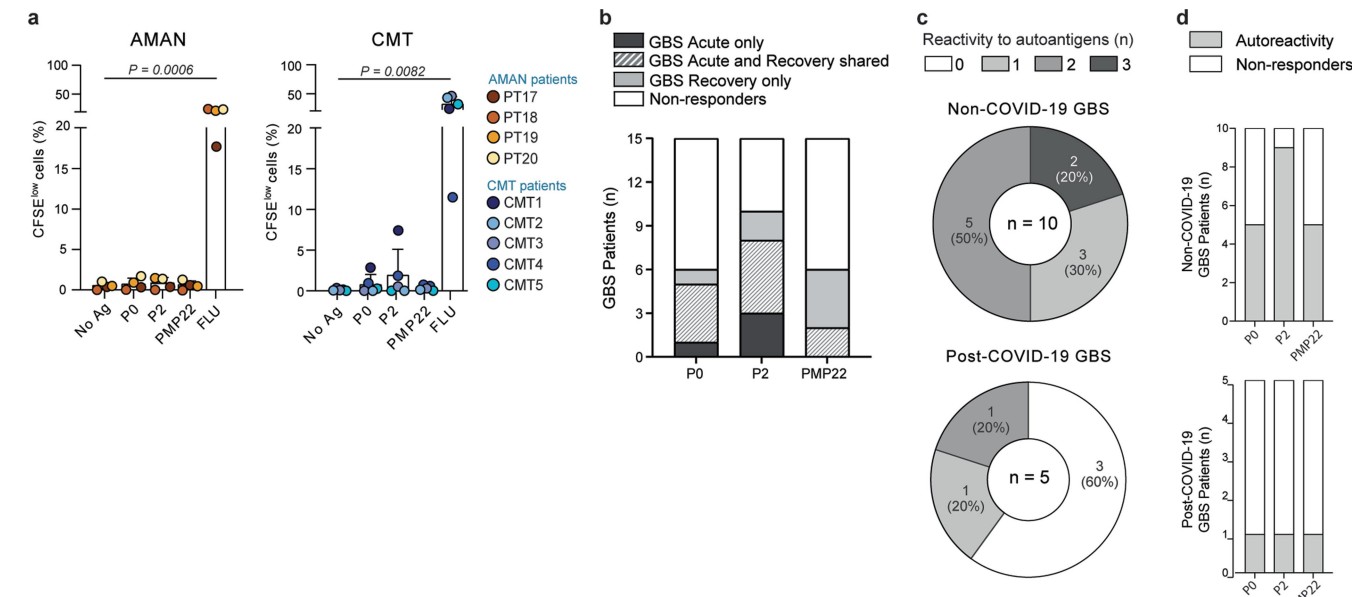

**Extended Data Fig. 2 | Overview of the autoreactive memory CD4⁺ T cell response in patients with GBS, with respect to disease subtype and stage and previous SARS-CoV-2 infection. a**, Scatter plot with pooled data from the indicated patients with AMAN (*n* = 4, biologically independent samples) and patients with CMT (*n* = 5 biologically independent samples) are shown as the percentage of proliferating CFSE^low cells. Each dot represents an individual donor and bar height indicates mean value with s.d. Data were analysed using two-tailed Wilcoxon matched-pairs signed rank test (*P* values are shown: **P < 0.01 and ***P < 0.001). **b**, Graphical representation of autoreactive response

detected on the basis of the total number of responding patients with GBS (*n* = 15, *y* axis) to P0, P2 and PMP22 myelin antigens (*x* axis) and disease phase in which autoreactivity was detected. **c**,**d**, Number and percentage of patients with GBS showing an autoreactive CD4⁺ T cell response divided with respect to previous SARS-CoV-2 infection (post-COVID-19, *n* = 5 and non-COVID-19, *n* = 10). **c**, Reactivity to one, two, three or none of the PNS-myelin autoantigens tested. **d**, Number of responding and non-responding patients with GBS shown with respect to P0, P2 and PMP22 myelin antigens.

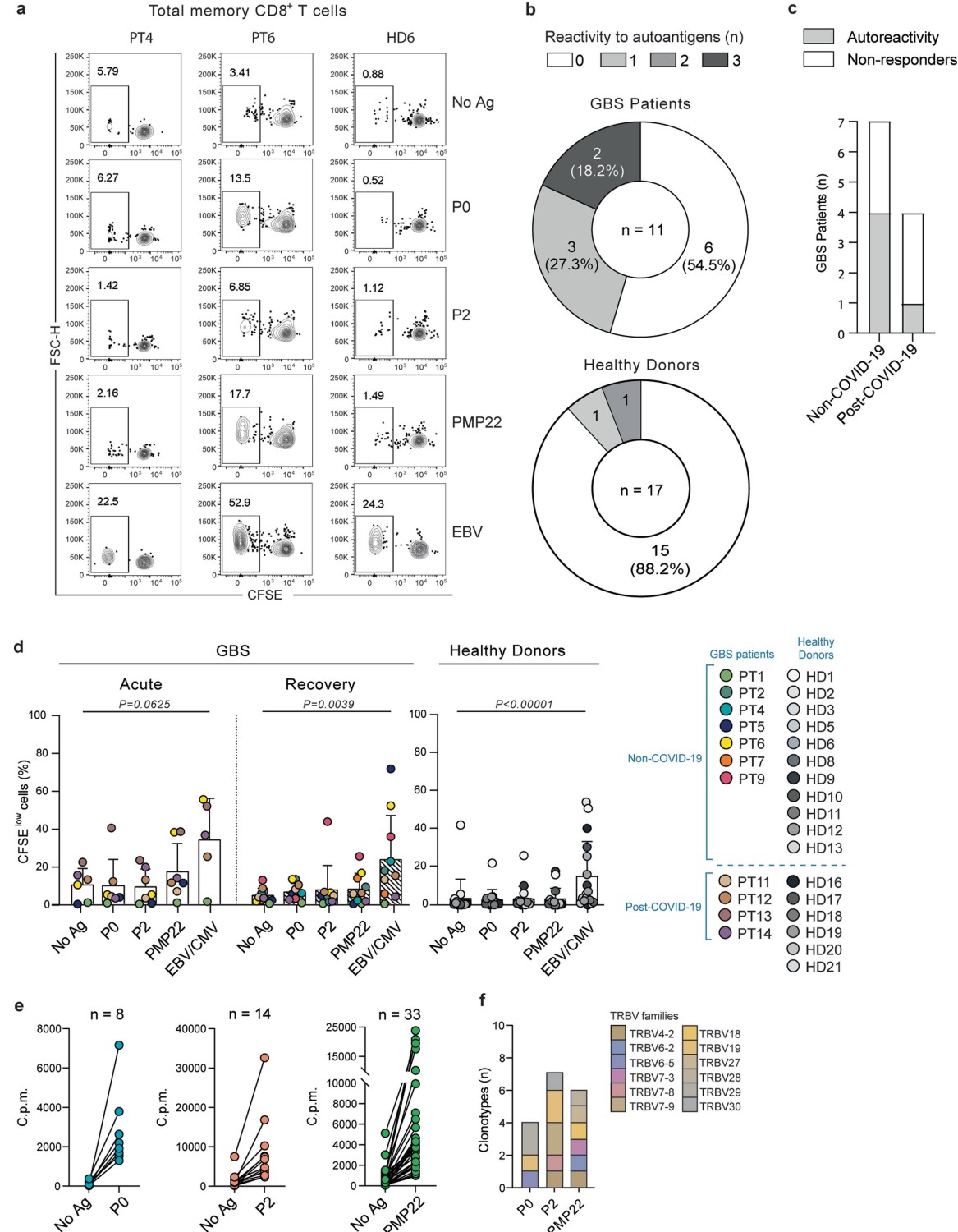

**Extended Data Fig. 3** | See next page for caption.

**Extended Data Fig. 3 | Ex vivo stimulation of memory CD8⁺ T cells from the blood of patients with GBS and healthy donors. a**, Total memory CD8⁺ T cells from the blood of patients with GBS and healthy donors were labelled with CFSE and cultured with autologous monocytes in the presence or absence of the PNS-myelin peptide pools (P0, P2, PMP22) and EBV/CMV HLA class I peptides as a positive control. CFSE profiles from two representative patients (PT4, PT6) and one healthy donor (HD6) are shown. **b**, Overview of the total number of screened patients with GBS and healthy donors reactive to one, two, three or none of the PNS-myelin autoantigens tested. Positive response was considered as stimulation index ≥ 2 and a Δ value ≥ 1.5%. **c**, Total number of patients with GBS showing, or not, an autoreactive CD8⁺ T cell response, divided with respect to previous SARS-CoV-2 infection. **d**, Scatter plot with pooled data from the patients with GBS (PT, $n = 11$ biologically independent samples, coloured dots) and healthy donors (HD, $n = 17$ biologically independent samples, white and grey dots),

shown as the percentage of proliferating CFSE^low cells. Each dot represents an individual donor and bar height indicates mean value with s.d. Data were analysed using two-tailed Wilcoxon matched-pairs signed rank test ($P$ values are shown). **e**, PNS-myelin specific memory CD8⁺ T cell clones were identified by individual stimulation with autologous B cells untreated (No Ag) or pulsed with the indicated PNS-myelin peptide pools (P0, P2, PMP22). Proliferation of CD8⁺ T cell clones was measured on day 3 after a 16-h pulse with [³H]-thymidine and is expressed as counts per minute (C.p.m.). Total number of PNS-myelin specific CD8⁺ T cell clones isolated from all patients with GBS ($n = 11$) is indicated above each graph. **f**, TCR Vβ gene usage of P0-, P2- and PMP22-specific CD8⁺ T cell clones isolated from patients with GBS ($n = 6$) is shown as a stacked bar chart. The $y$ axis indicates the number of unique autoreactive TCRβ clonotypes carrying different CDR3β sequences.

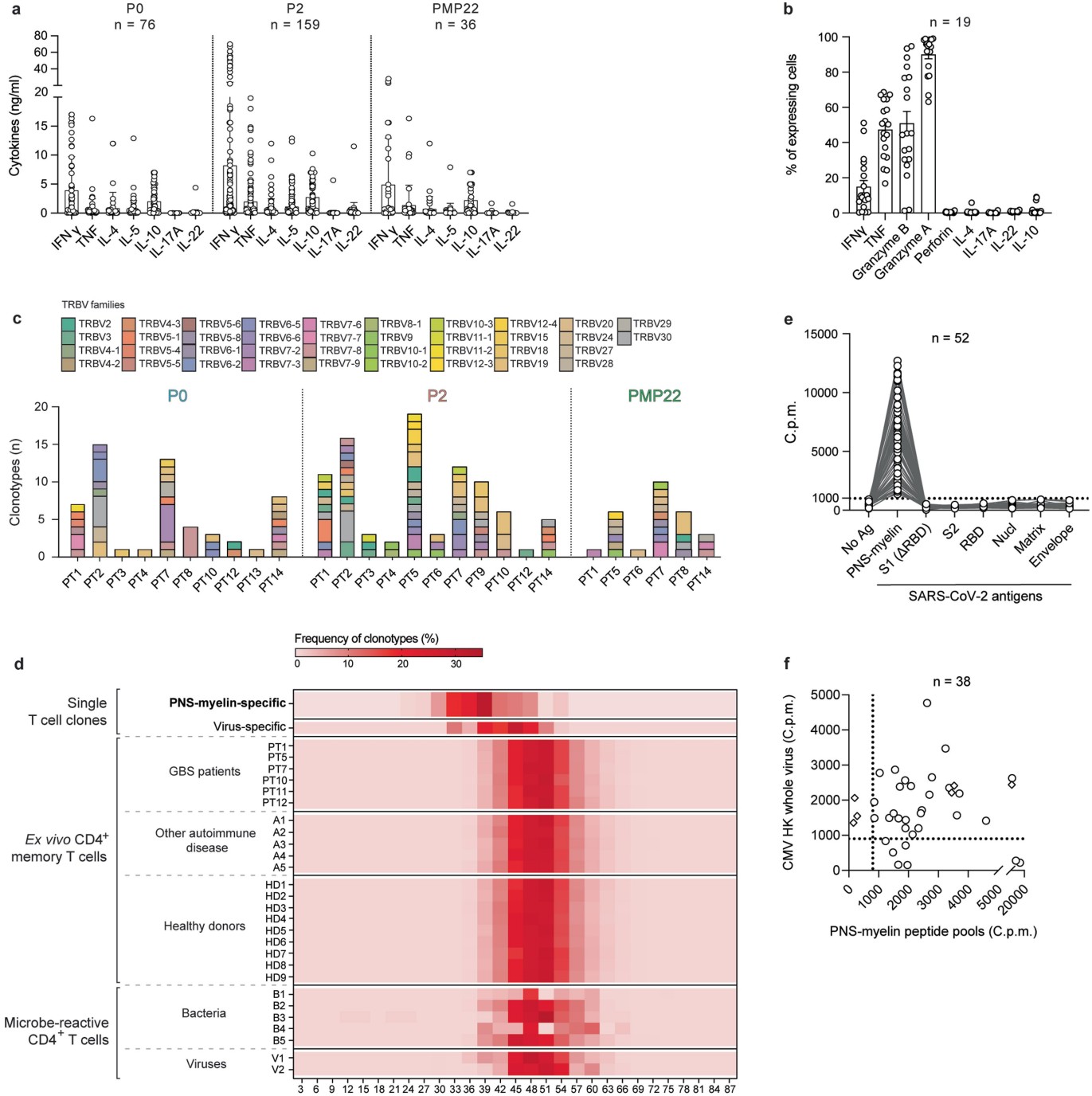

**Extended Data Fig. 4 | Characterization of autoreactive T cell clones in patients with GBS. a**,**b**, Cytokine production by P0-specific (*n* = 76 biologically independent samples), P2-specific (*n* = 159 biologically independent samples) and PMP22-specific (*n* = 36 biologically independent samples) CD4[+] T cell clones derived from patients with GBS analysed in the 48-h-culture supernatants after stimulation or not with the self-antigen by bead-based multiplex assay (**a**) or intracellularly by intracellular FACS staining (**b**). Each dot represents an individual clone and bar height indicates mean value with s.d. (**a**) or s.e.m. (**b**). **c**, TCR Vβ gene usage of P0-, P2- and PMP22-specific TCRβ clonotypes isolated from the indicated patients with GBS. The *y* axis indicates the number of autoreactive TCRβ clonotypes. **d**, Heat map showing the percentage of clonotypes bearing the same CDR3β length defined by the number of nucleotides. The CDR3β lengths of TCRβ clonotypes from PNS-myelin-specific CD4[+] T cells isolated from patients with GBS are shown in the top row (*n* = 166 from 13 PT) and were

compared to the CDR3β lengths of (i) SARS-CoV-2-specific CD4[+] T cell clones (virus-specific TCRβ clonotypes, *n* = 92 from 6 PT), (ii) ex vivo memory CD4[+] T cells from the indicated patients with GBS, or patients with other autoimmune disorders (A) or healthy donors (C) and (iv) bacteria- (B) and viruses- (V) reactive CD4[+] T cell CFSE[low] fractions enriched from in vitro stimulation assay. **e**,**f**, PNS-myelin specific CD4[+] T cell clones (circles) from two post-COVID-19 patients with GBS (*n* = 52; PT12 and PT14) or from CMV-associated patients with GBS (*n* = 32; PT2 and PT3) as well as CMV-specific CD4[+] T cell clones from PT2 (rhombus symbols, *n* = 6) were stimulated in the presence of irradiated autologous B cells pulsed with the PNS-myelin antigen or the indicated SARS-COV-2 antigens (**e**) or heat-inactivated human CMV (**f**). Shown are the C.p.m. values indicating the proliferation of autoreactive T cell clones measured after a 16-h pulse with [³H]-thymidine.

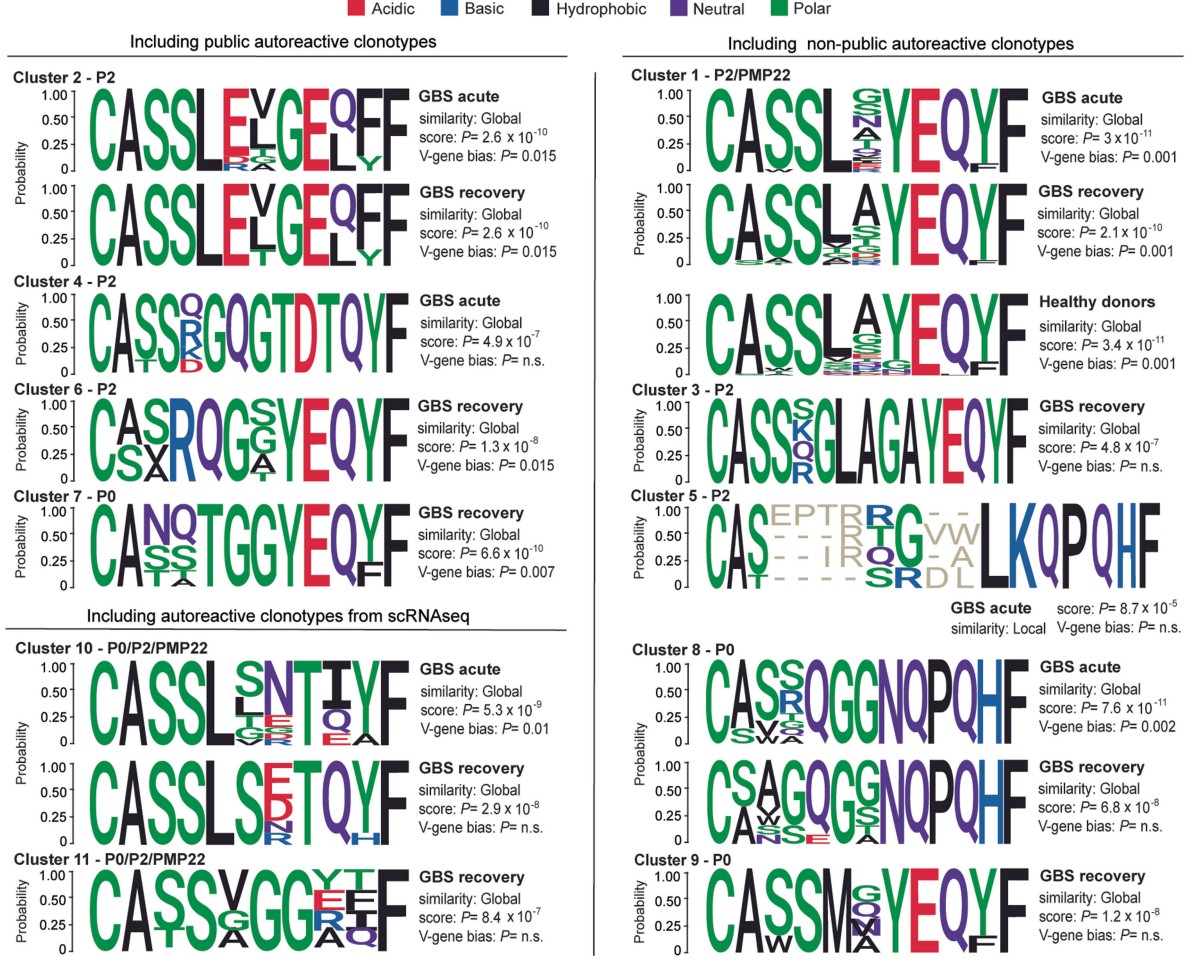

**Extended Data Fig. 5 | CDR3β consensus motifs characterizing the GLIPH2 specificity clusters.** Disease phase, similarity type, total GLIPH2 score and V-gene bias are shown for each cluster ($n$ = 11). Clusters including public and not public autoreactive TCRβ clonotypes in patients with GBS are indicated.

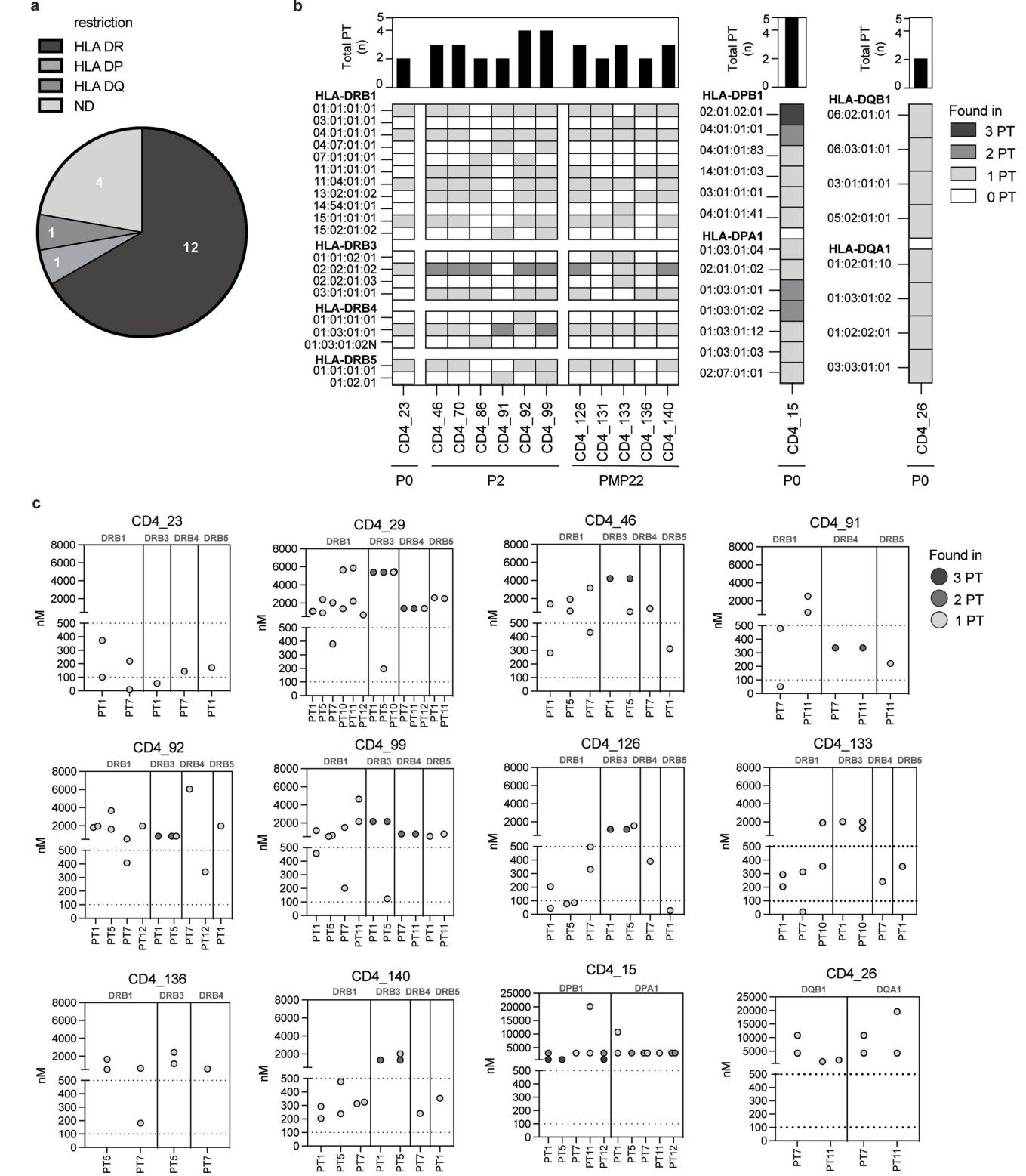

**Extended Data Fig. 6 | Association between HLA polymorphisms and public autoreactive TCR Vβ sequences in patients with GBS. a**, Pie chart indicating the distribution of the MHC restriction of the identified public TCRβ clonotypes (*n* = 18). **b**, Chart summarizing the total number of patients with GBS in whom each public TCRβ clonotype was found (top histograms); the HLA alleles (*y* axis) shared among those patients, with a colour-coded map indicating the number of patients sharing each allele; and the PNS-myelin antigen specificity of each public TCRβ clonotype (*x* axis). **c**, Dot plots indicating the NetMHCIIpan predicted affinity binding (nM) of the HLA alleles, from each patient carrying a public TCRβ clonotype, to their known cognate antigen, as determined by epitope mapping experiments. Dotted lines subdivide the graphs in high-affinity (lower part), middle-affinity, and low-affinity areas. The colour code indicates whether a specific allele was shared among patients with the same public TCRβ clonotype.

**Extended Data Table 1 | Patients included in this study**

| Patient ID | Gender | Age | Diagnosis | Prior infection | Treatment | Anti-gangliosides antibody (serum) | Acute blood sample* | Follow-up blood sample* | CSF sample | Nerve biopsy sample | GBS disability at follow up |
|---|---|---|---|---|---|---|---|---|---|---|---|
| PT1 | M | 74 | Non-COVID-19 AIDP | VZV | None | Negative | 7 | 201 | N.A. | N.A. | partial |
| PT2 | M | 31 | Non-COVID-19 AIDP | CMV | None | Anti-GD1a IgG Positive Anti-GD1b IgG Positive Anti-GM1 IgG Positive Anti-GM2 IgM Positive | 5 | 180 | N.A. | N.A. | full recovery |
| PT3 | M | 52 | Non-COVID-19 AIDP | CMV | IVIg | Negative | N.A. | 136 | N.A. | N.A. | partial |
| PT4 | M | 75 | Non-COVID-19 AIDP | N.A. | IVIg | Negative | 15 | 235 | N.A. | N.A. | partial |
| PT5 | F | 65 | Non-COVID-19 AIDP | N.A. | None | Negative | 36 | 314 | N.A. | N.A. | partial |
| PT6 | M | 30 | Non-COVID-19 AIDP | N.A. | None | Negative | 15 | 350 | N.A. | N.A. | partial |
| PT7 | M | 80 | Non-COVID-19 AIDP | N.A. | IVIg | N.A. | 0 | 223 | N.A. | N.A. | partial |
| PT8 | F | 61 | Non-COVID-19 AIDP | Pyelonephritis | IVIg | N.A. | 13 | 241 | N.A. | N.A. | partial |
| PT9 | M | 64 | Non-COVID-19 AIDP | N.A. | IVIg | Negative | 8 | 428 | N.A. | N.A. | full recovery |
| PT10 | F | 71 | Non-COVID-19 AIDP | N.A. | None | Negative | 4 | N.A. | Yes | N.A. | partial |
| PT11 | M | 54 | Post-COVID-19 AIDP | SARS-CoV-2 | IVIg and PEX | Negative | 20 | 135 | Yes | N.A. | partial |
| PT12 | M | 49 | Post-COVID-19 AIDP | SARS-CoV-2 | IVIg | Negative | 9 | 147 | Yes | N.A. | partial |
| PT13 | F | 48 | Post-COVID-19 AIDP | SARS-CoV-2 | IVIg | Negative | 9 | 509 | N.A. | N.A. | partial |
| PT14 | M | 49 | Post-COVID-19 AIDP | SARS-CoV-2 | None | Negative | 12 | 138 | N.A. | N.A. | partial |
| PT15 | M | 53 | Post-COVID-19 AIDP | SARS-CoV-2 | None | Anti-GM2 IgM Positive | 6 | 188 | N.A. | N.A. | full recovery |
| PT16 | M | 66 | Non-COVID-19 AIDP | VZV | PEX and IVIg | Negative | 31 | N.A. | N.A. | Yes | partial |
| PT17 | M | 64 | AMAN | N.A.§ | None | Anti-GD1a IgM Positive Anti-GD1a IgG Positive | 8 | N.A. | N.A. | N.A. | full recovery |
| PT18 | F | 67 | AMAN | N.A.§ | None | Anti-GM1 IgG Positive | 5 | N.A. | N.A. | N.A. | partial |
| PT19 | M | 65 | AMAN | N.A.§ | IVIg | Anti-GD1b IgG Positive | 4 | N.A. | N.A. | N.A. | partial |
| PT20 | M | 63 | AMAN | N.A.§ | None | N.A. | N.A. | 1520 | N.A. | N.A. | partial |
| CMT1 | F | 23 | CMT1A | N.A. | None | N.A. | N.A. | N.A. | N.A. | N.A. | N.A. |
| CMT2 | M | 31 | CMT1A | N.A. | None | N.A. | N.A. | N.A. | N.A. | N.A. | N.A. |
| CMT3 | F | 34 | CMT1A | N.A. | None | N.A. | N.A. | N.A. | N.A. | N.A. | N.A. |
| CMT4 | M | 45 | CMT1X | N.A. | None | N.A. | N.A. | N.A. | N.A. | N.A. | N.A. |
| CMT5 | M | 27 | CMT1X | N.A. | None | N.A. | N.A. | N.A. | N.A. | N.A. | N.A. |

N.A. not available; VZV, varicella-zoster virus; CMV, cytomegalovirus; IVIg, intravenous immunoglobulin; PEX, plasma exchange; *days from disease onset; §preceding gastroenteritis, specific microbial infection not defined.

**Extended Data Table 2 | List of experiments performed on samples from patients with AIDP**

| PT (ID) | Memory CD4+ T cells from blood | | | | Total CD4+ T cells from CFS | Memory CD8+ T cells from blood | | |
|---|---|---|---|---|---|---|---|---|
| | Ex vivo screening (disease stage) | PNS-myelin reactivity | Isolation of PNS-myelin specific clones | High Throughput TCRβ sequencing (disease stage) | High Throughput TCRβ sequencing (disease stage) | Ex vivo screening (disease stage) | PNS-myelin reactivity | Isolation of PNS-myelin specific clones |
| PT1 | AC+REC | yes | yes | AC + REC | N.D. | AC+REC | no | no |
| PT2 | AC+REC | yes | yes | AC + REC | N.D. | REC | no | no |
| PT3 | REC | yes | yes | N.D. | N.D. | N.D. | N.D. | N.D. |
| PT4 | AC+REC | yes | yes | REC | N.D. | REC | no | no |
| PT5 | AC+REC | yes | yes | AC + REC | N.D. | AC+REC | yes | yes |
| PT6 | AC+REC | yes | yes | N.D. | N.D. | AC+REC | yes | yes |
| PT7 | AC+REC | yes | yes | AC + REC | N.D. | REC | yes | yes |
| PT8 | AC+REC | yes | yes | N.D. | N.D. | N.D. | N.D. | N.D. |
| PT9 | AC+REC | yes | yes | REC | N.D. | REC | yes | yes |
| PT10 | AC | yes | yes | AC | yes | N.D. | N.D. | N.D. |
| PT11 | REC | no | no | AC | yes | AC+REC | yes | no |
| PT12 | AC+REC | yes | yes | AC + REC | yes | AC+REC | no | no |
| PT13 | AC | no | yes | REC | N.D. | AC | no | no |
| PT14 | AC+REC | yes | yes | N.D. | N.D. | AC+REC | no | yes |
| PT15 | AC+REC | no | no | N.D. | N.D. | N.D. | N.D. | N.D. |
| PT16 | AC (EM + CM) | yes | no | N.D. | N.D. | N.D. | N.D. | N.D. |

N.D.: not done.

|---|---|

# Reporting Summary

## Statistics

For all statistical analyses, confirm that the following items are present in the figure legend, table legend, main text, or Methods section.

| n/a | Confirmed | |
|---|---|---|
| ☐ | ☒ | The exact sample size (*n*) for each experimental group/condition, given as a discrete number and unit of measurement |
| ☐ | ☒ | A statement on whether measurements were taken from distinct samples or whether the same sample was measured repeatedly |
| ☐ | ☒ | The statistical test(s) used AND whether they are one- or two-sided<br>*Only common tests should be described solely by name; describe more complex techniques in the Methods section.* |
| ☒ | ☐ | A description of all covariates tested |
| ☒ | ☐ | A description of any assumptions or corrections, such as tests of normality and adjustment for multiple comparisons |
| ☐ | ☒ | A full description of the statistical parameters including central tendency (e.g. means) or other basic estimates (e.g. regression coefficient) AND variation (e.g. standard deviation) or associated estimates of uncertainty (e.g. confidence intervals) |
| ☐ | ☒ | For null hypothesis testing, the test statistic (e.g. *F*, *t*, *r*) with confidence intervals, effect sizes, degrees of freedom and *P* value noted<br>*Give P values as exact values whenever suitable.* |
| ☒ | ☐ | For Bayesian analysis, information on the choice of priors and Markov chain Monte Carlo settings |
| ☒ | ☐ | For hierarchical and complex designs, identification of the appropriate level for tests and full reporting of outcomes |
| ☒ | ☐ | Estimates of effect sizes (e.g. Cohen's *d*, Pearson's *r*), indicating how they were calculated |

*Our web collection on statistics for biologists contains articles on many of the points above.*

## Software and code

Policy information about availability of computer code

| Data collection | BD FACS Diva (v9.0) was used for acquisition of samples. The Microbeta2 Windows Workstation software (v2.3.0.12) was used to acquire data of T cell proliferation by [3H]-thymidine incorporation on a Beta counter 2 (Perkin Elmer). |
|---|---|
| Data analysis | IMGT (v3.5.31) was used to analyze TCR Vβ sequences obtained by Sanger method. ImmunoSEQ Analyzer V3.0 (http://www.immunoseq.com) was used for data processing of TCR Vβ CDR3 sequencing performed by Adaptive Biotechnologies. The R package immunarch 0.9.0 (https://github.com/immunomind/immunarch) was used to study antigen-specific clonotypes in each donor's repertoire according to bioidentity overlap, defined as identical identified V gene, amino acid sequence of the CDR3 region and identified J gene. The HetzDra/turboGliph 0.99.2 R package (https://github.com/HetzDra/turboGliph/) was used to group lymphocyte interaction by paratope hotspots and predict TCRβ specificity clusters. Graphpad Prism (v9) was used to analyze data and create plots. Seurat (v 4.9.9.9059) was used for the analysis of scRNAseq experiments. NetMHCIIpan 4.0 server was used for peptide-HLA affinity prediction. Data Analysis Software Suite for LEGENDplex™ (Biolegend) was used for the quantification cytokine release by autoreactive T cell clones. Flow-Jo (v10.8.1) was used for flow cytometry data analysis. Gene set enrichment analysis (GSEA) was performed using the software package "escape" (version 1.10.0, https://github.com/ncborcherding/escape). The R version used for all the computational analyis is v 4.2.1.. |

For manuscripts utilizing custom algorithms or software that are central to the research but not yet described in published literature, software must be made available to editors and reviewers. We strongly encourage code deposition in a community repository (e.g. GitHub). See the Nature Portfolio guidelines for submitting code & software for further information.

## Data

Policy information about availability of data

All manuscripts must include a data availability statement. This statement should provide the following information, where applicable:

- Accession codes, unique identifiers, or web links for publicly available datasets
- A description of any restrictions on data availability
- For clinical datasets or third party data, please ensure that the statement adheres to our policy

Publicly available datasets included in the study are available at the following web links: https://doi.org/10.21417/JSL2021S ; https://doi.org/10.21417/AC2020EJI and https://doi.org/10.21417/B73H0P, https://vdjdb.cdr3.net/search, https://www.ncbi.nlm.nih.gov/geo/query/acc.cgi (accession numbers GSE59114;, GSE126030, GSE131935, GSE104024, GSE193442) and https://ega-archive.org/ (accession numbers EGAS00001003215, EGAD00001005290. The data presented in this manuscript are tabulated in the main paper and in the supplementary materials. TCR Vβ sequences from samples listed in Extended Data Table 2 are deposited in the immuneACCESS database (https://www.immunoseq.com/immuneaccess/, DOI: https://doi.org/10.21417/LS2023N).

## Human research participants

Policy information about studies involving human research participants and Sex and Gender in Research.

| | |
|---|---|
| Reporting on sex and gender | The study cohort was comprised of both sexes as shown in Extended Data Table 1. Patients were included in the study as samples become available and sexes were included whenever possible. |
| Population characteristics | Relevant information on human research participants such as age, gender, disease, HLA-typing and anti-gangliosides antibody presence are provided in Extended Data Table 1 and Supplementary Table 3. Specifically, the study included 20 GBS patients (age range 30-50, 59.5 ± 13.3 (mean ± SD)) and 5 CMT1 patients (age range 23-45, 32 ± 8.4 (mean ± SD)) recruited from University Hospital Zurich and Cantonal Hospital of Lugano (EOC), and 21 HD obtained from the Swiss Blood Donation Center of Lugano (n = 15, age range 23-51, 47.5 ± 15.1 (mean ± SD)) and from the CoV-ETH study (n = 6, age range 38-50, 41.3 ± 4.6 (mean ± SD)). We included a total of 16 patients with AIDP, both at acute phase (range 7 - 36 days from disease onset, 12.7 ± 9.9 (mean ± SD)), and/or at follow-up visits during the recovery stage (range 135-509 days from disease onset, 244.6 ± 115.7 (mean ± SD)) (Extended Data Table 1). We also included AMAN patients (n = 4, all associated with preceding gastroenteritis) at disease onset (n=3, range 4-8 days from disease onset 5.7 ± 2.1 (mean ± SD) or recovery (n=1, 1520 days after disease onset), or CMT1 patients (n= 3 CMT1A and n=2 CMT1X). |
| Recruitment | Recruitment of patients was biased by the study design. Given the large clinical heterogeneity of GBS disease, the study specifically aimed at characterizing autoreactive T cell responses initially in matched acute/recovery samples from a defined subset of GBS patients, specifically suffering from the AIDP subtype. The AIDP patients were included in the study based on the diagnosis when matched acute/recovery samples become available. AMAN and CMT patients were included later based on the diagnosis as they become available. Diagnosis of AIDP, AMAN or CMT was based on the criteria of the National Institute of Neurological Disorders and Stroke (NINDS). Biological samples were collected at acute phase and/or at follow-up visits during the recovery stage (Extended Data Table 1). All patient samples were recruited from University Hospital Zurich and Cantonal Hospital of Lugano (EOC). Healthy donors were recruited at the Swiss Blood Donation Center of Lugano and post-COVID-19 healthy donors were recruited from the CoV-ETH cohort study and used as control group in all experiments. |
| Ethics oversight | The study was approved by the Ethical committees of Zurich (NeuroMyoCyTOF study, BASEC-Nr: 2016-00929; CoV-ETH cohort, BASEC-Nr. 2020-00949) and Lugano (IGOS study, BASEC-Nr: 2018-01860). All participants provided written informed consent for participation in the study. |

Note that full information on the approval of the study protocol must also be provided in the manuscript.

# Field-specific reporting

Please select the one below that is the best fit for your research. If you are not sure, read the appropriate sections before making your selection.

☒ Life sciences       ☐ Behavioural & social sciences       ☐ Ecological, evolutionary & environmental sciences

For a reference copy of the document with all sections, see nature.com/documents/nr-reporting-summary-flat.pdf

# Life sciences study design

All studies must disclose on these points even when the disclosure is negative.

| | |
|---|---|
| Sample size | AIDP (n=16), AMAN (n=4), CMT (n=5) patients and HD (n= 21). No statistical methods were used to predetermine sample size. GBS is a rare disease and we included in the study patients enrolled in different centers in Switzerland and based on the diagnosis of AIDP, AMAN or CMT subtype. Sample size was chosen based our previous studies (i.e. Narcolepsy study, Latorre et al, Nature, 2018) as well as on the clear evidence of significant enrichment of autoreactive T cells in GBS patients compared to healthy individuals. |
| Data exclusions | Data from bulk TCR sequencing of CD4+ memory T cells ex vivo from the blood of GBS patients were excluded from the calculation of the |

| Data exclusions | cumulative frequency (Fig 4d and h) if they had less than 5500 clonotypes (identified as productive rearrangements). In scRNAseq data analysis, on the basis of QC exclusion was done by filtering of low-quality cells and cell doublets or multiples, and cells with >5% mitochondrial counts, we normalized the data and performed scaling, dimensionality reduction and clustering on the top 2000 highly variable features in the dataset (Seurat v 4.9.9.9059). |
|---|---|
| Replication | In the cases of T cell clones, their reactivity against self-antigens was mostly assessed in multiple occasions (in at least 2 separate experiments) with reproducible results. |
| Randomization | Randomization was not possible given the experimental design. The AIDP patients were included in the study based on the diagnosis when matched acute/recovery samples become available. AMAN and CMT patients were included later based on the diagnosis. The requisite for inclusion in the study was based on the diagnosis of AIDP, AMAN and CMT according to the criteria of the National Institute of Neurological Disorders and Stroke (NINDS). |
| Blinding | Investigators were not blinded to allocation during experiments and outcome assessment due to the experimental design of the study and the low likelihood of bias in the particular experiments. Given the large clinical heterogeneity of GBS disease, the study specifically aimed at characterizing autoreactive T cell responses in matched acute/recovery samples from a defined subset of GBS patients, specifically suffering from the AIDP subtype. AMAN and CMT patients and post-COVID-19 healthy donors were included later in the study. The investigators were unaware of the initial infectious trigger of the disease in the case on AIDP and AMAN patients. Healthy donors were specifically included as control groups in all experiments. |

# Reporting for specific materials, systems and methods

We require information from authors about some types of materials, experimental systems and methods used in many studies. Here, indicate whether each material, system or method listed is relevant to your study. If you are not sure if a list item applies to your research, read the appropriate section before selecting a response.

## Materials & experimental systems

| n/a | Involved in the study |
|---|---|
| ☐ | ☒ Antibodies |
| ☒ | ☐ Eukaryotic cell lines |
| ☒ | ☐ Palaeontology and archaeology |
| ☒ | ☐ Animals and other organisms |
| ☒ | ☐ Clinical data |
| ☒ | ☐ Dual use research of concern |

## Methods

| n/a | Involved in the study |
|---|---|
| ☒ | ☐ ChIP-seq |
| ☐ | ☒ Flow cytometry |
| ☒ | ☐ MRI-based neuroimaging |

## Antibodies

| Antibodies used | CD3 BV785 Cat:300472; Clone:UCHT1; Biolegend <br> CD8 APC-Fire 750; Cat:301066; Clone:RPA-T8; Biolegend <br> CD8 FITC Cat: A07756 ; Clone: B9.11; Beckman Coulter <br> CD4 PE/Dazzle 594; Cat:300548; Clone:RPA-T4; Biolegend <br> CD45RA BV650 Cat:304136; Clone:HI100; Biolegend <br> CCR7 BV421 Cat:353208; Clone:G043H7; Biolegend <br> CD25 PC5 Cat:IM2646; Clone:B1.49.9; Beckman Coulter <br> CD56 PC5 Cat:A07789; Clone:N901; Beckman Coulter <br> CD14 PC5 Cat:A70204; Beckman Coulter <br> CD19 FITC Cat:555412; Clone:HIB19; BD Biosciences <br> ICOS Pacific Blue Cat:313522; Clone:H4A3; Biolegend <br> CD25 PE Cat:555432; Clone:M-A251; BD Biosciences <br> Granzyme A APC Cat:507220; Clone CB9; Biolegend <br> Granzyme B PE Cat:396406; Clone QA18A28; Biolegend <br> Perforin APC/Cyanine7 Cat:308128; Clone dG9; Biolegend <br> TNFα BV785 Cat:502948; Clone MAb11; Biolegend <br> IL-10 PE/Cyanine7 Cat:501404; Clone JES3-9D7; Biolegend <br> IL-17A BV605 Cat:512326; Clone BL168; Biolegend <br> IFNγ BUV395 Cat:563563; Clone B27; BD Biosciences <br> IL-4 BV711 Cat:564112; Clone MP4-25D2; BD Biosciences <br> IL-22 BUV737 Cat:367-7229-42; Clone 22URTI; Thermo Fisher Scientific <br><br> antiHLA-DR, clone L243, ATCC cat n HB55, produced in house from hybridoma cell line <br> anti-HLA-DQ clone SPVL3 (1), produced in house from hybridoma cell line <br> anti-HLA-DP, clone B7/21 (2), produced in house from hybridoma cell line <br><br> (1) H. Spits, G. Keizer, J. Borst, C. Terhorst, A. Hekman, J. E. de Vries, Characterization of monoclonal antibodies against cell surface molecules associated with cytotoxic activity of natural and activated killer cells and cloned CTL lines. Hybridoma 2, 423–437 (1983). doi:10.1089/hyb.1983.2.423 Medline <br> (2) A. J. Watson, R. DeMars, I. S. Trowbridge, F. H. Bach, Detection of a novel human class II HLA antigen. Nature 304, 358–361 (1983). doi:10.1038/304358a0 Medline |
|---|---|

| Validation | Commercial antibodies were used following the manufacturer's instructions. Anti-HLA neutralizing antibodies were validated by detecting expected blocking of proliferation on T cell clones with known HLA restriction.

CD3 BV785 Cat:300472; Clone:UCHT1; Biolegend: https://www.biolegend.com/fr-ch/products/brilliant-violet-785-anti-human-cd3-antibody-14454
CD8 APC-Fire 750; Cat:301066; Clone:RPA-T8; Biolegend: https://www.biolegend.com/fr-ch/products/apc-fire-750-anti-human-cd8a-antibody-13580
CD8 FITC Cat: A07756 ; Clone: B9.11; Beckman Coulter: https://www.beckman.de/reagents/coulter-flow-cytometry/antibodies-and-kits/single-color-antibodies/cd8/A07756
CD4 PE/Dazzle 594; Cat:300548; Clone:RPA-T4; Biolegend:  https://www.biolegend.com/fr-ch/products/pe-dazzle-594-anti-human-cd4-antibody-9780
CD45RA BV650 Cat:304136; Clone:HI100; Biolegend: https://www.biolegend.com/fr-ch/products/brilliant-violet-650-anti-human-cd45ra-antibody-7662
CCR7 BV421 Cat:353208; Clone:G043H7; Biolegend: https://www.biolegend.com/fr-ch/products/brilliant-violet-421-anti-human-cd197-ccr7-antibody-7497
CD25 PC5 Cat:IM2646; Clone:B1.49.9; Beckman Coulter: https://www.beckman.de/reagents/coulter-flow-cytometry/antibodies-and-kits/single-color-antibodies/cd25/IM2646
CD56 PC5 Cat:A07789; Clone:N901; Beckman Coulter: https://www.beckman.es/reagents/coulter-flow-cytometry/antibodies-and-kits/single-color-antibodies/cd56/A07789
CD14 PC5 Cat:A70204; Beckman Coulter: https://www.beckman.com/reagents/coulter-flow-cytometry/antibodies-and-kits/single-color-antibodies/cd14/a70204
CD19 FITC Cat:555412; Clone:HIB19; BD Biosciences: https://www.bdbiosciences.com/en-ch/products/reagents/flow-cytometry-reagents/research-reagents/single-color-antibodies-ruo/fitc-mouse-anti-human-cd19.555412
ICOS Pacific Blue Cat:313522; Clone:H4A3; Biolegend: https://www.biolegend.com/fr-ch/products/pacific-blue-anti-human-mouse-rat-cd278-icos-antibody-7373
CD25 PE Cat:555432; Clone:M-A251; BD Biosciences: https://www.bdbiosciences.com/en-ch/products/reagents/flow-cytometry-reagents/research-reagents/single-color-antibodies-ruo/pe-mouse-anti-human-cd25.555432
Granzyme A APC Cat:507220; Clone CB9; Biolegend: https://www.biolegend.com/fr-ch/products/apc-anti-human-granzyme-a-antibody-15038
Granzyme B PE Cat:396406; Clone QA18A28; Biolegend: https://www.biolegend.com/fr-ch/products/pe-anti-human-mouse-granzyme-b-recombinant-antibody-17396
Perforin APC/Cyanine7 Cat:308128; Clone dG9; Biolegend: https://www.biolegend.com/fr-ch/products/apc-cyanine7-anti-human-perforin-antibody-13027
TNFα BV785 Cat:502948; Clone MAb11; Biolegend: https://www.biolegend.com/fr-ch/products/brilliant-violet-785-anti-human-tnf-alpha-antibody-12027
IL-10 PE/Cyanine7 Cat:501404; Clone JES3-9D7; Biolegend: https://www.biolegend.com/fr-ch/products/pe-anti-human-il-10-antibody-1341
IL-17A BV605 Cat:512326; Clone BL168; Biolegend: https://www.biolegend.com/fr-ch/products/brilliant-violet-605-anti-human-il-17a-antibody-7879
IFNγ BUV395 Cat:563563; Clone B27; BD Biosciences: https://www.bdbiosciences.com/en-ch/products/reagents/flow-cytometry-reagents/research-reagents/single-color-antibodies-ruo/buv395-mouse-anti-human-ifn.563563
IL-4 BV711 Cat:564112; Clone MP4-25D2; BD Biosciences: https://www.bdbiosciences.com/en-ch/products/reagents/flow-cytometry-reagents/research-reagents/single-color-antibodies-ruo/bv711-rat-anti-human-il-4.564112
IL-22 BUV737 Cat:367-7229-42; Clone 22URTI; Thermo Fisher Scientific: https://www.thermofisher.com/antibody/product/IL-22-Antibody-clone-22URTI-Monoclonal/367-7229-42 |

# Flow Cytometry

## Plots

Confirm that:

☒ The axis labels state the marker and fluorochrome used (e.g. CD4-FITC).

☒ The axis scales are clearly visible. Include numbers along axes only for bottom left plot of group (a 'group' is an analysis of identical markers).

☒ All plots are contour plots with outliers or pseudocolor plots.

☒ A numerical value for number of cells or percentage (with statistics) is provided.

## Methodology

| Sample preparation | PBMCs were isolated with Ficoll-Paque Plus (GE Healthcare). Monocytes were enriched by positive selection using CD14-coated microbeads (Miltenyi Biotec).  From the CD14– cell fraction, memory CD4+ and CD8+ total cells were sorted to over 98% purity on a FACSAria Fusion (BD) excluding CCR7+CD45RA+, CD25bright, CD14+, CD56+ cells as well as either CD8+ cells (for memory CD4+ T cell enrichment) or CD4+ cells (for memory CD8+ T cell enrichment) according to the gating strategy shown in Extended Data Figure 1a. For intracellular cytokine staining, clones were restimulated with Phorbol-12-myristat-13-acetat (PMA) and Ionomycin in the presence of brefeldin A (all from Sigma-Aldrich), fixed and permeabilized with Cytofix/Cytoperm (BD Biosciences) according to the manufacturer's instructions. |

| Instrument | BD FACSAria Fusion
BD LSRFortessa |

| Software | BD FACS Diva (v9.0) was used for acquisition of samples and Flow-Jo (v10.8.1) for analsyis. |
|---|---|
| Cell population abundance | Purity of the relevant cell populations (CD4+ and CD8+ memory T cells) was checked after sorting at BD LSRFortessa instrument and found to be >98%. |
| Gating strategy | CD4+ and CD8+ total memory cells were sorted from CD14- fractions with BD FACSAria Fusion instrument excluding excluding CCR7+CD45RA+ double positive, CD25bright, CD14+ and CD56+ cells s well as either CD8+ cells (for memory CD4+ T cell enrichment) or CD4+ cells (for memory CD8+ T cell enrichment) according to the gating strategy shown in Extended Data Figure 1a. CD4+ T cells from CSF and nerve biopsy were sorted from T cell lines expanded in vitro using a BD FACSAria Fusion instrument as CD3+CD4+CD8-–CD19–CD56– . |

☒ Tick this box to confirm that a figure exemplifying the gating strategy is provided in the Supplementary Information.

