## [Peer Review File · Nature]

Manuscript Title: Autoreactive T cells target peripheral nerves in Guillain-Barré Syndrome.

Reviewer Comments & Author Rebuttals

Reviewer Reports on the Initial Version:

Referees' comments:

Referee #1 (Remarks to the Author):

This is a technically well conducted and interesting study demonstrating the presence of autoreactive myelin-specific T cells (mainly CD4) in patients who suffer from GBS after episodes of viral infection (SARS-CoV-2) or viral reactivation (VZV and CMV).

The authors utilized a "mixture" of methods to analyse T cells: T cell screening through a proliferative assay against myelin peptide pools; T cell cloning and TCR sequencing to show the presence of myelin-reactive T cells. The authors determine fine-specificity and HLA-restriction of such myelin-reactive T cells that were detected mostly in GBS (12/15) and rarely (1/15) in Healthy controls. The authors also showed that these T cells can be identify by the presence of public and private TCR clonotypes and they detected such T cells also in some nerve biopsies and in the CSF.

The data collected support therefore the conclusion of the authors. The data presented in the manuscript demonstrate the presence of autoreactive memory CD4+ T cells in the blood of GBS patients, both at disease onset and recovery. As such, this study represents a very important demonstration that such autoreactive T cells can have an important contribution to the pathogenesis of GBS.

Major

a) The detailed analysis of the specificity of T cells detected in GBS patients contrast however with the absolute lack of any functional characterization of such T cells. We don't have any idea of whether such autoreactive T cells present a peculiar cytokine production profile (IL-17?, TNF-alpha?) or if they can be directly cytotoxic. The complete lack of functional analysis represents in my opinion an important weakness of this interesting manuscript, but I also think that the authors can provide a direct functional characterization of such T cells or derive their possible function through analysis of their gene-expression profile.

b) Since the majority of the autoreactive myelin-specific T cells are CD4 and HLA-Dr restricted, the question is whether such T cells recognize myelin epitopes directly presented at the surface of the nerves or by antigen-presenting cells present in close proximity. If not experimentally, the authors should at least discuss whether neural cells can express MHC-class II (in physiological or under inflammatory conditions). This other part of the puzzle is also completely ignored in the paper discussion.

Minor

a) The GBS patients studied suffered different viral infections before disease (Varicella-zoster virus (VZV) (n = 2), Cytomegalovirus (CMV) (n = 2) or SARS-CoV-2 (n=5). Since the authors characterize the fine specificity of a large number of T cell clones, I am wondering whether there were different

patterns of recognition related to previous infection and if some possible “cross-reactivity” related to shared viral sequences could be detected in the different patients in relation to the apparent triggering viral infection.

Referee #2 (Remarks to the Author):

In this submission titled “T cells targeting peripheral nerves are expanded and shared in patients with Guillain-Barre Syndrome,” Sukenikova et al. provide evidence for myelin-specific autoreactive T cells being enriched in GBS patients. The authors isolate CD4+ T cells initially from the blood of patients and show that some of these cells respond to myelin antigens. Through the use of TCRb sequencing, the authors further observed that these myelin-specific T cells bore TCRb chains with shorter CDR3s than those from viral-specific T cells. Lastly, the authors also observe these autoreactive cells in the nerves and CSF of patients. Overall, the authors provide the first evidence for autoreactive T cells possibly being causative in GBS.

Major points:

- 1) It is curious that the authors did not assess the functional phenotypes of any of their clones/isolated T cells. Are these myelin-specific T cells producing IFNg? IL-17A? Cytolytic molecules? Other cytokines/chemokines? What are the surface markers expressed by some of these cells?
- 2) It is important to note that the majority of the responses are directed against HLA-DR. HLA-DR is an interesting MHC-II molecule because the majority of the polymorphisms in this heterodimer have been linked to the beta chain with few polymorphisms being observed in the alpha chain. Since the TCRb chain tends to interact more with the MHC-II alpha chain, is that the reason why the authors observe sharing of TCR Vb chains and CDR3b sequences? Are the shared TCRb chains also HLA-DR-restricted? Do patients with shared CDR3b sequences in their autoreactive T cells also share MHC-II alleles?
- 3) The authors observe that the autoreactive T cells tend to express TCRs with shorter CDR3 regions. Shorter CDR3s have been previously linked to degenerate peptide responses (PMID: 8777724). Do the authors observe something similar with these T cells?
- 4) Hydrophobic doublet residues in the P6-P7 positions of the CDR3b have been reported to enhance autoreactivities in T cells (PMID: 27348411). In Fig. 3e, it appears that polar/charged residues are more apparent at these positions in the autoreactive T cells. How do the authors reconcile their observations with what has been previously published? Giving some context to their observations will make the paper more interesting and relevant.
- 5) In Fig. 4a, when the authors isolate T cells from the CSF, why are there such few autoreactive T cells (as determined by the TCRb sequences)? Based on the authors' claims, these cells should be expanded, especially in GBS patients. It is surprising that “such clonotypes were amongst the lowest expanded.”
- 6) Since the authors know both the epitopes and the HLA molecules that autoreactive T cells tend to recognize, this allows them to generate pMHC multimers (tetramers/dextramers) so they can track these cells using their specificities without having to isolate them from a bulk population based on responses following in vitro activation (which could distort any pre-existing functional biases). Doing

so would allow for more in-depth analyses such as phenotype, transcriptomics, and also performing paired TCRa/TCRb sequencing using scRNA-seq since currently the authors have only sequenced the TCRb chains. Altogether, this would be a valuable tool to generate and provide frequency analysis and function of these potential autoreactive clones which would enhance the overall message in this submission.

Minor points:

1) Although the authors are generally careful with their wording regarding the TCR repertoire sharing, it is critical that the authors employ the same care throughout the manuscript. For example, the title of the manuscript refers to T cells being shared across GBS patients. This is a misleading claim for multiple reasons – 1) T cells are not shared but rather the TCR repertoire is shared, 2) the authors did not demonstrate TCR sharing but rather TCRb sharing since the TCRa sequence for each cell remains unknown. A more accurate claim would be that the antigenic specificities are shared across multiple patients. A similar misleading title is used as a section header for Figure 3 where the authors say “TCR repertoire analysis identifies persistent and public autoreactive T cells in GBS patients.” Again, the repertoire can be public but the T cells themselves are not public. Within an individual, two T cells bearing identical TCRs may have distinct phenotypes/functions so it is entirely possible that T cells across individuals may also be phenotypically/functionally different. “Public T cells” connotes that there is sharing at the cellular level instead of the repertoire level, which would be an inaccurate/misleading claim without substantial data to support it. This also applies to when the authors use the word clonotypes. Although they are once again careful, there are times in the manuscript when they write clonotypes instead of TCRb clonotypes.

2) In the methods section, the authors write that to isolate memory CD4+ and CD8+ T cells, “memory CD4+ and CD8+ total cells were sorted to over 98% purity on a FACSAria Fusion (BD) excluding CD45RA+, CD25bright, CD8+ or CD4+, CD14+ and CD56+ cells.” This statement is unclear because it reads as if the authors are excluding CD8+ and CD4+ T cells but presumably that is not the intent. This statement needs to be clarified and likely amended.

3) While the authors include in their methods section that they exclude CD25bright cells to isolate the memory CD4+ T cells, no gating strategy is included to show how this sort was performed. This would be critical to include in a manuscript. Presumably the CD25bright T cell exclusion was to ensure that Tregs were not being sorted for downstream analyses. Including Tregs in the experiments would certainly obfuscate the overall conclusions because Tregs are known to be self-reactive. But how can the authors more comprehensively ensure that Tregs are excluded? Could CD127 be used as well?

In summary, although the authors provide some interesting primary evidence for the existence and role of autoreactive CD4+ T cells in GBS pathology, the study in its current form is incomplete with many unanswered questions. The message would be dramatically improved and the study would generate more excitement if repertoire analysis, with function analysis was done together with the patient's disease status.

Referee #3 (Remarks to the Author):

This study is a highly detailed account of the characteristics of the T cell repertoire reactive with PO, P2 and/or PMP-22 in the AIDP variant of GBS relative to healthy controls. This typical finding for EAN has never been successfully translated into human GBS, so represents an important step forward. Whilst not completely original in scope and aim, it is nevertheless the most comprehensive and insightful research effort in this area to reach publication stage. For many years autoreactivity to these myelin proteins has been considered central to AIDP pathogenesis, yet high quality positive evidence has been lacking. This study provides such data.

The laboratory conducting this work has deep expertise in the area of T cell cloning and characterisation that appears to have been thoroughly and carefully conducted consistent with their prior studies (eg on narcolepsy). I am not sufficiently fluent in the technical approach to make a judgment on quality, yet on the face of it the work seems well conducted.

The sample size for GBS (n=16) and controls (n=15) is reasonable, considering the amount of work required to process each individual case. The results reach statistical significance and indicate rather generally that there is an expanded CD4 T cell repertoire in this disorder. If this is a correct and reproducible result it is a highly significant step forward in our understanding of the disease.

The manuscript is clear and well written and of interest to general researchers in autoimmunity, as well as subject-specific researchers.

Whilst the experimental T cell work seems to me highly excellent (as far as my limited expertise can tell), there are some areas of the sampling and additional considerations that could be further developed in future work.

1. The use of healthy controls as opposed to post-infectious controls (CMV, VZV, Covid etc) is a sub-optimal choice. It would be reassuring to know that equivalent post-infectious controls (ie. without GBS) performed in a similar way to healthy controls.
2. An attempt to seek and find autoantibody responses to these myelin proteins and match them with CD4 T cell data would provide an additional layer of evidence around causality. Other have looked for this and not found them. Perhaps the group have looked but not declared the data as it is negative. If so, I would encourage them to do so.
3. A comparison of the T cell repertoire with cases of C. jejuni-associated GBS in which (T cell independent) anti-ganglioside antibodies are the known pathogenic factor would provide important additional data on causality. i.e. if T cell expansion to myelin antigens was found it would suggest these are a secondary phenomenon. An alternative disease category would be the Miller Fisher variant of GBS where anti-GQ1b ganglioside antibody is the casual factor.
4. Another excellent control group would be the genetically-determined demyelinating neuropathies (CMT1) that are referred to in the text.
5. A single nerve biopsy, whilst revealing an important finding, is a modest number and could be strengthened by increasing the n.
6. Extending the myelin protein screening to inclusion of the CIDP-associated paranodal antigens (Caspr, contacting NF155/186 etc) or unbiased whole myelin screening would be worthwhile.
7. In normal clinical practice it would be odd to see 16 cases of AIDP without any cases of AMAN,

which makes me wonder about the clinical sampling bias and how the inclusion and exclusion criteria were handled. This section could be expanded.

All these additional points would help to shore up the robustness of the finding with respect to causality. Notwithstanding this, the study looks excellent and important as a step forward in its current form.

We thank the Reviewers for their positive comments and constructive criticisms. Based on their suggestions, we have revised our manuscript and included new data in the revised Figures 1 (panels b, c) and in the new Figure 2 (panels a-e), 3 (panels a, c-f) and 4 (panels a-h) as well as in Extended Data Figures 1 (panels a,b), 2 (panel a), 3 (panels a,b,d), 4 (panels a-f) and 5 (panels a-c). Additional information has been added in Supplementary Tables 1, 2 and 4 as well as in Extended Data Table 1 and 2. Below is a point by- point reply to the Reviewers' comments.

Point-by-point reply to the Referees' comments

Referee #1 (Remarks to the Author):

This is a technically well conducted and interesting study demonstrating the presence of autoreactive myelin-specific T cells (mainly CD4) in patients who suffer from GBS after episodes of viral infection (SARS-CoV-2) or viral reactivation (VZV and CMV).

The authors utilized a "mixture" of methods to analyse T cells: T cell screening through a proliferative assay against myelin peptide pools; T cell cloning and TCR sequencing to show the presence of myelin-reactive T cells. The authors determine fine-specificity and HLA-restriction of such myelin-reactive T cells that were detected mostly in GBS (12/15) and rarely (1/15) in Healthy controls. The authors also showed that these T cells can be identify by the presence of public and private TCR clonotypes and they detected such T cells also in some nerve biopsies and in the CSF.

The data collected support therefore the conclusion of the authors. The data presented in the manuscript demonstrate the presence of autoreactive memory CD4⁺ T cells in the blood of GBS patients, both at disease onset and recovery. As such, this study represents a very important demonstration that such autoreactive T cells can have an important contribution to the pathogenesis of GBS.

We thank the Reviewer for appreciating our work and for the specific comments.

Major

a) The detailed analysis of the specificity of T cells detected in GBS patients contrast however with the absolute lack of any functional characterization of such T cells. We don't have any idea of whether such autoreactive T cells present a peculiar cytokine production profile (IL-17?, TNF-alpha?) or if they can be directly cytotoxic. The complete lack of functional analysis represents in my opinion an important weakness of this interesting manuscript, but I also think that the authors can provide a direct functional characterization of such T cells or derive their possible function through analysis of their gene-expression profile.

We thank the Reviewer for raising this important issue, which was raised also by the other Reviewers. We followed the Reviewer's suggestion and analysed a panel of autoreactive CD4⁺ T cell clones for their cytokine production in the 48h-culture supernatants upon stimulation or not with the self-antigen by bead-based multiplex assay (LEGENDplex™ HU Th Cytokine Panel, BioLegend) as well as for their intracellular cytokine expression by flow cytometry. The results of this analysis, which have been added in Extended Figure 4a and b, point to a cytotoxic T helper 1 (Th1)-like phenotype of autoreactive CD4⁺ T cells in GBS patients.

Moreover, we optimized an approach based on the combination of our *in vitro* screening assays with scRNAseq analysis to gain further information on the potential phenotype of autoreactive T cells. This study was performed on memory CD4⁺ T cells from 2 GBS patients (PT2 and PT16) at day 6 from *in vitro* stimulation with PNS-myelin or Flu antigens in the presence of autologous monocytes. When compared to Flu-reactive T cells, self-reactive T cells showed a similar Th1-like phenotype, but higher cytotoxic signature, confirming our data from single T cell clone analysis. Notably, gene set enrichment analysis identified high enrichment scores within a gene set that was previously associated to autoimmune diseases (MeSH ID:D001327) in self-reactive T cells, but not in Flu-reactive T cells (MeSH ID:D007251). The results of this study are now included in a new Figure 2.

b) Since the majority of the autoreactive myelin-specific T cells are CD4 and HLA-DR restricted, the question is whether such T cells recognize myelin epitopes directly presented at the surface of the nerves or by antigen-presenting cells present in close proximity. If not experimentally, the authors should at least discuss whether neural cells can express MHC-class II (in physiological or under inflammatory conditions). This other part of the puzzle is also completely ignored in the paper discussion.

We thank the Reviewer for raising this important point. We now refer in the discussion (lines 479-484) to the possibility that myelin-specific CD4⁺ T cells may see the cognate antigens locally on infiltrating macrophages (PMID: 11240209) as well as directly on Schwann cells, which have previously been reported to show increased antigen presenting capacity and MHC class II molecules expression under inflammatory conditions (PMID: 19544394, PMID: 28970572, PMID: 19914379).

Minor

a) The GBS patients studied suffered different viral infections before disease (*Varicella-zoster virus (VZV)* ($n = 2$), *Cytomegalovirus (CMV)* ($n = 2$) or *SARS-CoV-2* ($n=5$). Since the authors characterize the fine specificity of a large number of T cell clones, I am wondering whether there were different patterns of recognition related to previous infection and if some possible “cross-reactivity” related to shared viral sequences could be detected in the different patients in relation to the apparent triggering viral infection.

We thank the Reviewer for raising the point of potential cross-reactivity between self-antigens and viral antigens of autoreactive clones from viral infections-associated GBS cases. No significant shared viral sequences could be detected by Basic Local Alignment Search Tool (BLAST) analysis (data not shown) nor specific epitope recognition patterns of T clone reactivity was observed when analysed in relation to previous viral infection, such as VSV-associated PT1; SARS-CoV-2-associated PT12, PT13 and PT14; HCMV-associated PT2 and PT3, which is now described in the text and shown here in Figure R1 for the Reviewers).

However, in view of a possible functional cross-reactivity, we screened autoreactive T cell clones derived from post-COVID-19 GBS patients ($n= 59$; PT12 and PT14) or HCMV-associated GBS patients ($n = 32$; PT2 and PT3) respectively against SARS-CoV-2 antigens or HCMV. While none of the PNS-myelin reactive clones from post-COVID-19 GBS patients showed cross-reactivity to SARS-COV-2 antigens, the majority of autoreactive clones (26 out of 32 clones) isolated from HCMV-associated GBS patients showed proliferation in response to heat inactivated HCMV. We also obtained 6 HCMV-specific clones from PT2 and, when screened for cross-reactivity, 3 of them showed proliferation against both P0 and P2 antigens. These data are now included in Extended Data Figure 4e and f and Supplementary Table 1. Follow up studies will further characterize such cross-reactive clones to define the epitope recognized and the degree of cross-reactivity with a larger panel of self and microbial antigens.

Overall, these findings suggest that certain preceding infectious agents may be directly involved in the disease establishment by inducing self-reactive T cell immunity in GBS. Further analysis of cross-reactive T cell immunity should be performed on more GBS patients associated with distinct preceding infectious agents to increase our understanding of the undelaying cellular and molecular mechanisms.

Figure R1

Referee #2 (Remarks to the Author):

In this submission titled “T cells targeting peripheral nerves are expanded and shared in patients with Guillain-Barre Syndrome,” Sukenikova et al. provide evidence for myelin-specific autoreactive T cells being enriched in GBS patients. The authors isolate CD4+ T cells initially from the blood of patients and show that some of these cells respond to myelin antigens. Through the use of TCRb sequencing, the authors further observed that these myelin-specific T cells bore TCRb chains with shorter CDR3s than those from viral-specific T cells. Lastly, the authors also observe these autoreactive cells in the nerves and CSF of patients. Overall, the authors provide the first evidence for autoreactive T cells possibly being causative in GBS.

In summary, although the authors provide some interesting primary evidence for the existence and role of autoreactive CD4+ T cells in GBS pathology, the study in its current form is incomplete with many unanswered questions. The message would be dramatically improved and the study would generate more excitement if repertoire analysis, with function analysis was done together with the patient's disease status.

We thank the Reviewer for appreciating our work and for the raising important points.

Major points:

1) It is curious that the authors did not assess the functional phenotypes of any of their clones/isolated T cells. Are these myelin-specific T cells producing IFNg? IL-17A? Cytolytic molecules? Other cytokines/chemokines?

We thank the Reviewer for raising this important issue, which was raised also by the other Reviewers. We followed the Reviewer's suggestion and analysed a panel of autoreactive CD4⁺ T cell clones for their cytokine production in the 48h-culture supernatants upon stimulation or not with the self-antigen by bead-based multiplex assay (LEGENDplex™ HU Th Cytokine Panel, BioLegend) as well as for their intracellular cytokine expression by flow cytometry. The results of this analysis, which have been added in Extended Figure 4a and b, point to a cytotoxic T helper 1 (Th1)-like phenotype of autoreactive CD4⁺ T cells in GBS patients.

Moreover, we optimized an approach based on the combination of our *in vitro* screening assays with scRNAseq analysis to gain further information on the potential phenotype of autoreactive T cells. This study was performed on memory CD4⁺ T cells from 2 GBS patients (PT2 and PT16) at day 6 from *in vitro* stimulation with PNS-myelin or Flu antigens in the presence of autologous monocytes. When compared to Flu-reactive T cells, self-reactive T cells showed a similar Th1-like phenotype, but higher cytotoxic signature, confirming our data from single T cell clone analysis. Notably, gene set enrichment analysis identified high enrichment scores within a gene set that was previously associated to autoimmune diseases (MeSH ID:D001327) in self-reactive T cells, but not in Flu-reactive T cells (MeSH ID:D007251). The results of this study are now included in a new Figure 2.

2) It is important to note that the majority of the responses are directed against HLA-DR. HLA-DR is an interesting MHC-II molecule because the majority of the polymorphisms in this heterodimer have been linked to the beta chain with few polymorphisms being observed in the alpha chain. Since the TCRb chain tends to interact more with the MHC-II alpha chain, is that the reason why the authors observe sharing of TCR Vb chains and CDR3b sequences? Are the shared TCRb chains also HLA-DR-restricted? Do patients with shared CDR3b sequences in their autoreactive T cells also share MHC-II alleles?

We appreciate the issue raised here on the role of HLA polymorphisms in influencing the presence of shared TCRβ sequences across GBS patients. To address this point, we performed a more comprehensive analysis showing that shared GBS TCRβ clonotypes were mostly HLA-DR restricted (n = 12 out of 18) and only two were either HLA-DP or HLA-DQ restricted, whereas for 4 of them we were not able to determine their HLA restriction. We then analysed each of the shared GBS TCRβ clonotype in relation to its HLA restriction and the HLA class II alleles carried by GBS patients from whom that specific clonotype was identified. Focusing on HLA-DR-restricted public TCRβ clonotypes, this analysis did not identify any bias in HLA-DRB1 allele sharing across patients. However, HLA-DRB3 02:02:01:02 allele was found shared by 2 patients in the case of 5 P2-specific clonotypes (out of 6) and of 2 PMP22-specific clonotypes (out of 5). Furthermore, HLA-DRB4 01:03:01:01 allele was shared by 2 patients in the case of 2 P2-specific clonotypes (out of 6). Although these data point to a potential role for HLA-DRB3/4 in shaping the autoreactive TCR clonotype repertoire in GBS patients, further studies on a larger cohort of patients should be performed to draw any significant conclusions. Additionally, we performed NetMHCIIpan prediction of binding affinity of the cognate epitope for each public GBS TCRβ clonotype in relation to the HLA alleles of the patients from whom that clonotype was identified. This analysis revealed that distinct HLA class II alleles are predicted to bind their cognate peptides within a similar range of affinities across patients, suggesting a promiscuous binding of peptides by HLA class II allele variants. The results of this analysis are now included in the new Extended Data Figure 5 and in the text under the new paragraph entitled “Promiscuous binding of self-peptides with similar affinity by distinct HLA class II alleles” (lines 307-330).

3) The authors observe that the autoreactive T cells tend to express TCRs with shorter CDR3 regions. Shorter CDR3s have been previously linked to degenerate peptide responses (PMID: 8777724). Do the authors observe something similar with these T cells?

We agree with the Reviewer that shorter CDR3 regions have been previously linked to degenerate peptide responses, although this aspect remains to be fully elucidated. While we have not specifically investigated this issue in the current study, we observed that in a few cases the same TCR V β clonotype showed reactivity against both P2 and P0 (Clonotype ID: CD4_19 and CD4_43) or PMP22 (Clonotype ID: CD4_136) as well as, in other cases, T cell clones displayed reactivity against either one or 2 distinct PNS-myelin antigens and HCMV (Clonotype ID: CD4_167) (Supplementary Table 1). These findings suggest the potential involvement of TCR degeneracy. We plan to perform a follow up work aiming at exploring the extent of TCR degeneracy of PNS-myelin reactive T cells in GBS patients, especially in the context of cross-reactivity between self and microbial antigens. Shedding light on this aspect may provide important insights into the disease pathophysiology. We have discussed this aspect and quoted the suggested paper in the discussion (lines 434-448).

4) Hydrophobic doublet residues in the P6-P7 positions of the CDR3 β have been reported to enhance autoreactivities in T cells (PMID: 27348411). In Fig. 3e, it appears that polar/charged residues are more apparent at these positions in the autoreactive T cells. How do the authors reconcile their observations with what has been previously published? Giving some context to their observations will make the paper more interesting and relevant.

The study cited by the Reviewer indicates that interfacial hydrophobicity of CDR3 β P6-P7 doublets plays a role in influencing both the frequency and strength of TCR self-reactivity regardless of the V β family or CDR3 length in normal mice and mice prone to type 1 diabetes. However, it should be noted that in the same study the authors indicated that their findings do not entirely align with previous observations in the experimental allergic encephalomyelitis (EAE) mouse model. In this model, self-reactive CD4⁺ T cells and Treg cells specific for myelin oligodendrocyte glycoprotein (MOG) did not show such biased usage of CDR3 β P6-P7 doublets (PMID: 20005134).

To address the Reviewer's comment, we conducted an analysis of the CDR3 β P6-P7 doublets among all PNS-myelin specific clonotypes isolated from GBS patients (Supplementary Table 1). Additionally, we examined self-reactive CD4⁺ T cell clonotypes previously identified in patients with narcolepsy, type 1 diabetes, and Rasmussen encephalitis (PMID: 30232458; PMID: 29259996; PMID: 25302633; PMID: 16380476; PMID: 27942583; PMID: 25681349; PMID: 19535636; PMID: 11978633; PMID: 12606513; PMID: 25157096; PMID: 15277377; PMID: 20299476). The results of this comprehensive analysis revealed that the hydrophobic doublet residues in the CDR3 β P6-P7 positions are not consistently present among autoreactive CD4⁺ T cells in various autoimmune disease contexts. This suggests that the antigen specificity of the T cell response may exert a more dominant influence over the repertoire-wide selection biases highlighted in the study referenced by the Reviewer. We have included these findings in Figure R2 for the Reviewers and they can be included as supplemental data if so required.

Figure R2

5) In Fig. 4a, when the authors isolate T cells from the CSF, why are there such few autoreactive T cells (as determined by the TCR β sequences)? Based on the authors' claims, these cells should be expanded, especially in GBS patients. It is surprising that "such clonotypes were amongst the lowest expanded."

We thank the Reviewer for this comment. It is important to note that, in our study, CSF-infiltrating T cells were polyclonally expanded *in vitro* with PHA and irradiated allogeneic feeder cells and IL-2 (500 IU/ml) from a limited CSF sample size (1-2 ml), which allowed us to perform TCR V β repertoire characterization and to search for the potential existence of PNS-myelin-reactive T cells. Despite the limited number of total clonotypes that could be retrieved from each sample (n = 552 in PT10; n = 3812 in PT1, and n = 892 in PT12), this methodology successfully identified certain autoreactive TCR V β clonotypes, some of them being shared across patients, thereby revealing their presence in the CSF. However, we are unable to infer whether the low expansion of these autoreactive clonotypes reflects their frequency *in vivo* or rather signifies a diminished capacity of such cells to be robustly expanded by our *in vitro* polyclonal expansion approach, which could potentially be attributed to an effector-like phenotype of such cells.

6) Since the authors know both the epitopes and the HLA molecules that autoreactive T cells tend to recognize, this allows them to generate pMHC multimers (tetramers/dextramers) so they can track these cells using their specificities

without having to isolate them from a bulk population based on responses following *in vitro* activation (which could distort any pre-existing functional biases). Doing so would allow for more in-depth analyses such as phenotype, transcriptomics, and also performing paired TCR α /TCR β sequencing using scRNA-seq since currently the authors have only sequenced the TCR β chains. Altogether, this would be a valuable tool to generate and provide frequency analysis and function of these potential autoreactive clones which would enhance the overall message in this submission.

We thank the Reviewer for this suggestion. We thoroughly considered the potential of investigating autoreactive T cells directly *ex vivo* from the blood of GBS patients, thereby circumventing the need for *in vitro* stimulation. However, this approach encounters significant challenges due to the considerable heterogeneity in the HLA landscape across GBS patients (Supplementary Table 3 and Extended Data Figure 5), the relatively low frequency of *ex vivo* autoreactive clones in the blood, and the limited availability of PBMCs in our laboratory. In light of these challenges, we pursued an alternative strategy. Specifically, we performed scRNAseq analysis and paired TCR V α / β sequencing on bulk memory CD4⁺ T cells derived from PT2 and PT16 at day 6 after *in vitro* stimulation either with a mixture of P0, P2 and PMP22 antigens (PNS-myelin antigens) or with Flu vaccine. This analysis was able to distinguish between antigen-reactive and non-reactive cells in response to stimulation with PNS-myelin or Flu antigens. When compared to Flu-reactive T cells, self-reactive T cells exhibited a comparable T helper 1 (Th1)-like phenotype yet displayed a more pronounced cytotoxic signature confirming our data from single T cell clone analysis. Notably, gene set enrichment analysis identified high enrichment scores within a gene set that was previously associated to autoimmune conditions in self-reactive T cells (MeSH ID:D001327), but not in Flu-reactive T cells (MeSH ID:D007251).

However, this approach did not allow us to identify any distinctive gene signatures in self-reactive T cells when analyzed in relation to the disease state (Figure R3a and b). Despite the expression of some genes (i.e., *IFNG*, *CCL4*, *CCL3*, *CCL5*, *FGFBP2*, *NKG7*, and *HOPX*) being higher in PNS-myelin reactive cells from the recovery phase compared to acute one, this analysis may be biased by technical limitations related to the inclusion of paired acute/recovery samples from only one GBS patient (PT2) as well as to the low cell number of antigen-reactive T cells retrieved from the recovery phase (from PNS-myelin stimulation condition: PT2 acute, n = 688; PT2 recovery, n = 262; PT16 acute, n = 1110; from Flu stimulation: PT2 acute, n = 287; PT2 recovery, n = 224; PT16 acute, n = 1721). We also observed similar patterns in acute and recovery phases when performing the gene set enrichment analysis using a gene set that was previously associated to autoimmune condition (MeSH ID:D001327) and Influenza infection (MeSH ID:D007251) (Figure R4c). Due to the technical limitations discussed above and the similarity in gene expression profiles of autoreactive T cells between disease states, we performed the analysis on combined acute and recovery datasets, as described above and in the manuscript (lines 133-174).

Overall, our data indicate that, despite the short *in vitro* activation, virus and self-reactive T cells in GBS patients display distinct T cell signatures. The results of this study are now included in a new Figure 2.

Figure R3

Minor points:

1) Although the authors are generally careful with their wording regarding the TCR repertoire sharing, it is critical that the authors employ the same care throughout the manuscript. For example, the title of the manuscript refers to T cells being shared across GBS patients. This is a misleading claim for multiple reasons – 1) T cells are not shared but rather the TCR repertoire is shared, 2) the authors did not demonstrate TCR sharing but rather TCRb sharing since the TCRA sequence for each cell remains unknown. A more accurate claim would be that the antigenic specificities are shared across multiple patients. A similar misleading title is used as a section header for Figure 3 where the authors say “TCR repertoire analysis identifies persistent and public autoreactive T cells in GBS patients.” Again, the repertoire can be public but the T cells themselves are not public. Within an individual, two T cells bearing identical TCRs may have distinct phenotypes/functions so it is entirely possible that T cells across individuals may also be phenotypically/functionally different. “Public T cells” connotes that there is sharing at the cellular level instead of the repertoire level, which would be an inaccurate/misleading claim without substantial data to support it. This also applies to when the authors use the word clonotypes. Although they are once again careful, there are times in the manuscript when they write clonotypes instead of TCRb clonotypes.

We thank the Reviewer for pointing out these errors, which have been corrected in the revised manuscript. In addition, the title of the manuscript has been revised.

2) In the methods section, the authors write that to isolate memory CD4+ and CD8+ T cells, “memory CD4+ and CD8+ total cells were sorted to over 98% purity on a FACSAria Fusion (BD) excluding CD45RA+, CD25bright, CD8+ or CD4+, CD14+ and CD56+ cells.” This statement is unclear because it reads as if the authors are excluding CD8+ and CD4+ T cells but presumably that is not the intent. This statement needs to be clarified and likely amended.

As suggested, we have revised the mentioned statement in the method section.

3) While the authors include in their methods section that they exclude CD25bright cells to isolate the memory CD4+ T cells, no gating strategy is included to show how this sort was performed. This would be critical to include in a manuscript. Presumably the CD25bright T cell exclusion was to ensure that Tregs were not being sorted for downstream analyses. Including Tregs in the experiments would certainly obfuscate the overall conclusions because Tregs are known to be self-reactive. But how can the authors more comprehensively ensure that Tregs are excluded? Could CD127 be used as well?

Tregs were excluded by sorting Dump- cells and excluding Dump+ cells that included (CD25^{bright} cells as well as CD56⁺ and CD14⁺ cells) according to the gating strategy that we have now included in Extended Data Figure 1b. In addition, as suggested by the Reviewer we have included CD127 staining in our flow cytometry panel, as shown in Figure R4 for the Reviewer, confirming the exclusion of Tregs (CD25 high CD127-) by using the gating strategy performed in the study.

Figure R4

Referee #3 (Remarks to the Author):

This study is a highly detailed account of the characteristics of the T cell repertoire reactive with P0, P2 and/or PMP-22 in the AIDP variant of GBS relative to healthy controls. This typical finding for EAN has never been successfully translated into human GBS, so represents an important step forward. Whilst not completely original in scope and aim, it is nevertheless the most comprehensive and insightful research effort in this area to reach publication stage. For many years autoreactivity to these myelin proteins has been considered central to AIDP pathogenesis, yet high quality positive evidence has been lacking. This study provides such data.

The laboratory conducting this work has deep expertise in the area of T cell cloning and characterisation that appears to have been thoroughly and carefully conducted consistent with their prior studies (eg on narcolepsy). I am not sufficiently fluent in the technical approach to make a judgment on quality, yet on the face of it the work seems well conducted.

The sample size for GBS (n=16) and controls (n=15) is reasonable, considering the amount of work required to process

each individual case. The results reach statistical significance and indicate rather generally that there is an expanded CD4 T cell repertoire in this disorder. If this is a correct and reproducible result it is a highly significant step forward in our understanding of the disease.

The manuscript is clear and well written and of interest to general researchers in autoimmunity, as well as subject-specific researchers.

Whilst the experimental T cell work seems to me highly excellent (as far as my limited expertise can tell), there are some areas of the sampling and additional considerations that could be further developed in future work.

We thank the Reviewer for appreciating our work.

1. The use of healthy controls as opposed to post-infectious controls (CMV, VZV, Covid etc) is a sub-optimal choice. It would be reassuring to know that equivalent post-infectious controls (ie. without GBS) performed in a similar way to healthy controls.

We thank the Reviewer for the suggestion. We have performed *in vitro* stimulation assay on total memory CD4⁺ and CD8⁺ T cell populations from the blood of 6 post-COVID-19 healthy controls at 2-3 weeks after SARS-CoV-2 infection. The results of this analysis, which are now included in Figure 1b and c as well as in the discussion, indicate that autoreactivity against PNS-myelin antigens and high background proliferation in negative control cultures is not detected in post-infectious controls, but it seems to be rather a peculiarity of post-COVID-19 GBS patients. These findings encourage further studies aiming at understanding autoreactive T cell immunity in patients with COVID-19-associated neurological disorders.

2. An attempt to seek and find autoantibody responses to these myelin proteins and match them with CD4 T cell data would provide an additional layer of evidence around causality. Other have looked for this and not found them. Perhaps the group have looked but not declared the data as it is negative. If so, I would encourage them to do so.

We agree with the reviewer about the relevance of screening for the presence of autoantibody against PNS-myelin proteins in our cohort of GBS patients. We attempted to perform ELISA screenings for the presence of IgG antibodies against P0, P2 or PMP22 proteins in the serum of all GBS patients and healthy donors included in our study. However, we faced some technical limitations associated with high background signal, which prevented us to draw any meaningful conclusions. In a follow up study, we plan to optimize ELISA assay for the detection of serum antibodies against P0, P2 or PMP22 antigens to address this important aspect.

3. A comparison of the T cell repertoire with cases of C. jejuni-associated GBS in which (T cell independent) anti-ganglioside antibodies are the known pathogenic factor would provide important additional data on causality. i.e. if T cell expansion to myelin antigens was found it would suggest these are a secondary phenomenon. An alternative disease category would be the Miller Fisher variant of GBS where anti-GQ1b ganglioside antibody is the casual factor.

We thank the Reviewer for this suggestion. We have performed *in vitro* stimulation assay on total memory CD4⁺ and CD8⁺ T cells from the blood of 4 AMAN patients (n = 3 at acute disease phase and n = 1 during the recovery phase) associated with preceding gastroenteritis (Extended Data Table 1). The results of this analysis, which are now shown in Extended Data Figure 2a, point to the absence of PNS-myelin reactive CD4⁺ T cells in such patients, suggesting that autoreactive T cell response targeting PNS-myelin proteins may have distinct roles in the demyelinating forms of GBS compared to the axonal disease variants.

4. Another excellent control group would be the genetically-determined demyelinating neuropathies (CMT1) that are referred to in the text.

We thank the Reviewer for this suggestion. We have performed *in vitro* stimulation assay on total memory CD4⁺ T cells from the blood of 5 CMT patients suffering from the demyelinating CMT1A and CMTX1 subtypes. Detailed information about these patients is included in Extended Data Table 1. PNS-myelin reactive CD4⁺ T cells were detected only in 1 out of 5 patients, suggesting that an autoreactive T cell immunity is mostly absent in CMT patients, but may potentially contribute to the disease in some sporadic cases. The results of this analysis are now included in Extended Data Figure 2a.

5. A single nerve biopsy, whilst revealing an important finding, is a modest number and could be strengthened by increasing the n.

We agree with the Reviewer's comment that analysing T cell infiltrates from a higher number of patients' nerve biopsies would be ideal. However, nerve biopsy is very rarely performed in GBS patients and it is not part of the routine diagnostic tests, which prevents us to have access to more samples. However, we believe that the data, in spite of being obtained from only 1 GBS patient, provides proof of principle of the ability of myelin-specific T cells to infiltrate the affected

tissue, thus strongly supporting their direct contribution to the disease immunopathology and therefore providing important information for a better understanding of GBS pathophysiology.

6. Extending the myelin protein screening to inclusion of the CIDP-associated paranodal antigens (Caspr, contacting NF155/186 etc) or unbiased whole myelin screening would be worthwhile.

We agree with the Reviewer's suggestion that it would be important to extend the analysis to the CIDP disease, which however is very rare and heterogeneous. We are indeed currently enrolling CIDP patients and we will be able to perform such analysis and report in future studies.

7. In normal clinical practice it would be odd to see 16 cases of AIDP without any cases of AMAN, which makes me wonder about the clinical sampling bias and how the inclusion and exclusion criteria were handled. This section could be expanded.

As suggested by the Reviewer we have clarified this point in the Method section (lines 621-624).

All these additional points would help to shore up the robustness of the finding with respect to causality. Notwithstanding this, the study looks excellent and important as a step forward in its current form.

Reviewer Reports on the First Revision:

Referees' comments:

Referee #1 (Remarks to the Author):

The authors addressed my criticisms. They added information about T cell function (figure 4a,b) and discussed the issue of possible limited antigen presentation of cells of neural origin. I am fully satisfied by the addition of these data. Congratulations.

Referee #2 (Remarks to the Author):

In this resubmission by Sukenikova et al. titled "Autoreactive T Cells targeting peripheral nerves in patients with Guillain-Barre Syndrome are expanded and share TCRb clonotypes", the authors demonstrate that autoreactive CD4+ T cells can be found in the CSF of GBS patients and are specific for myelin antigens. These cells bear a Th1 functional phenotype and express TCRs with short CDR3b chains that display a preferential restriction for HLA-DR MHC-II alleles. Overall, this is an interesting study that helps to provide great insight into neuropathies. The authors have addressed all of my concerns.

Referee #3 (Remarks to the Author):

The authors have addressed my comments satisfactorily. Whilst there is still much to be discovered and further clarified, the data do appear to support a role for myelin protein autoreactive T cells in some cases of post-viral demyelinating GBS - the subtype of GBS in which least is known about antigen targets and the T/B cell autoimmune repertoire.

It appears that the authors may have found a T cell repertoire associated with "post-viral GBS" which (as they point out) is most commonly the sensorimotor AIDP form of the disease. In C. jejuni associated pure motor GBS (often called "AMAN" that accounts for ~25-60% of GBS cases depending on geography) overwhelming evidence in the vast majority of cases points to an entirely different mechanism (T cell independent -TI- humoral immunity). I also suspect that the majority of URTI infections that trigger GBS and Miller Fisher syndrome (MFS) in which TI anti-ganglioside (GQ1b) antibodies are found (and are sufficiently and solely causal) are bacterial infections (mycoplasma/haemophilus etc) rather than viral infections. I would thus encourage the authors to be more cautious and clearer on these points as they may regret making over general claims about the generality of their findings to GBS in the future. Whilst I recognise that the title needs to be plain and short, they should at least in the abstract include that they are (mainly and where known) looking at post-viral AIDP and that in contrast they have not found similar T cell profiles in C. jejuni AMAN. Indeed both CZV and CMV are very (very!) rarely found as infectious drivers of GBS - only found in a few percentage of cases (see for eg the latest paper on preceding infections from IGOS by Leonhard S et al), the commonest known infection by far worldwide being C jejuni.

As I said in my last review I would really encourage them to look also at MFS in future studies. The inclusion of CMT as a control is good - but they should do more in future as 1/5 (20%) is a significant finding in my view. The sample size is still pretty small considering the huge heterogeneity of GBS.

The lack of identification of autoantibodies to these myelin proteins could be related to TH1/ vs TH2, so they may indeed be absent rather than difficult to find - a brief remark in the discussion about this.

Minor points: the AMAN patients mentioned around line ~103 should be formally included in the text about clinical sampling around line ~93.

Author Rebuttals to First Revision:

Nature manuscript 2023-03-04386A

We thank the Reviewers for their positive comments. Below is a point-by-point reply to the Reviewer #3's last comments.

Point-by-point reply to the Referees' comments

Referee #3 (Remarks to the Author):

The authors have addressed my comments satisfactorily. Whilst there is still much to be discovered and further clarified, the data do appear to support a role for myelin protein autoreactive T cells in some cases of post-viral demyelinating GBS - the subtype of GBS in which least is known about antigen targets and the T/B cell autoimmune repertoire.

We thank the Reviewer for appreciating our work and for the specific comments.

It appears that the authors may have found a T cell repertoire associated with "post-viral GBS" which (as they point out) is most commonly the sensorimotor AIDP form of the disease. In C. jejuni associated pure motor GBS (often called "AMAN" that accounts for ~25-60% of GBS cases depending on geography) overwhelming evidence in the vast majority of cases points to an entirely different mechanism (T cell independent -TI- humoral immunity). I also suspect that the majority of URTI infections that trigger GBS and Miller Fisher syndrome (MFS) in which TI anti-ganglioside (GQ1b) antibodies are found (and are sufficiently and solely causal) are bacterial infections (mycoplasma/haemophilus etc) rather than viral infections. I would thus encourage the authors to be more cautious and clearer on these points as they may regret making over general claims about the generality of their findings to GBS in the future. Whilst I recognise that the title needs to be plain and short, they should at least in the abstract include that they are (mainly and where known) looking at post-viral AIDP and that in contrast they have not found similar T cell profiles in C. jejuni AMAN. Indeed both CZV and CMV are very (very!) rarely found as infectious drivers of GBS - only found in a few percentage of cases (see for eg the latest paper on preceding infections from IGOS by Leonhard S et al), the commonest known infection by far worldwide being C jejuni.

We thank the reviewer for raising this important issue and we agree with the suggestion of being more cautious on making general statements. Indeed, our study has primarily focused on a homogenous group of GBS patients with demyelinating disease (with a fraction of them being post-viral) and European genetic background. We also agree that, despite having included a limited number of AMAN patients, which are much less frequent in our European geographical area and mostly associated to preceding bacterial infections, our data strongly point to the existence of distinct immune mediated mechanisms underlying AIDP versus AMAN cases. We modified the abstract accordingly (lines 25-26 and 33) and, when necessary, the text through the manuscript to describe our GBS patients as AIDP.

As I said in my last review I would really encourage them to look also at MFS in future studies. The inclusion of CMT as a control is good - but they should do more in future as 1/5 (20%) is a significant finding in my view. The sample size is still pretty small considering the huge heterogeneity of GBS.

We are aware of the limited sample size and of the importance to further characterize autoreactive T cell immunity in distinct GBS subtypes as well as in CMT patients. We are grateful to the reviewer for encouraging the inclusion of AMAN and CMT patients during the revision process, which has certainly improved the quality and the robustness of our findings and opened new scenarios to future studies. However, we agree that further investigations on a larger cohort of such patients, also including MFS, is needed to shed light on the immune mechanisms underlying distinct diseases subtypes. We added some comments on this issue in the discussion (lines 317-319).

The lack of identification of autoantibodies to these myelin proteins could be related to TH1/ vs TH2, so they may indeed be absent rather than difficult to find - a brief remark in the discussion about this.

As suggested, we have revised the part of the discussion section referring to the inconsistency in current data in identifying autoantibodies against PNS-myelin antigens in GBS patients (lines 356-358).

Minor points: the AMAN patients mentioned around line ~103 should be formally included in the text about clinical sampling around line ~93.

We thank the Reviewer for the accuracy in his revision. We modified the text accordingly.